



Earth System
Dynamics

# Relating climate sensitivity indices
# to projection uncertainty

**Benjamin Sanderson**[1,2]

[1]CERFACS, Toulouse, France
[2]NCAR, Boulder, CO, USA

**Correspondence:** Benjamin Sanderson (sanderson@cerfacs.fr)

**Abstract.** Can we summarize uncertainties in global response to greenhouse gas forcing with a single number? Here, we assess the degree to which traditional metrics are related to future warming indices using an ensemble of simple climate models together with results from the Coupled Model Intercomparison Project phases 5 and 6 (CMIP5 and CMIP6). We consider effective climate sensitivity (EffCS), transient climate response (TCR) at $CO_2$ quadrupling (T140) and a proposed simple metric of temperature change 140 years after a quadrupling of carbon dioxide (A140). In a perfectly equilibrated model, future temperatures under RCP8.5 (Representative Concentration Pathway 8.5) are almost perfectly described by T140, whereas in a mitigation scenario such as RCP2.6, both EffCS and T140 are found to be poor predictors of 21st century warming, and future temperatures are better correlated with A140. We show further that T140 and EffCS calculated in full CMIP simulations are subject to errors arising from control model drift and internal variability, with greater relative errors in estimation for T140. As such, if starting from a non-equilibrated state, measured values of effective climate sensitivity can be better correlated with true TCR than measured values of TCR itself. We propose that this could be an explanatory factor in the previously noted surprising result that EffCS is a better predictor than TCR of future transient warming under RCP8.5.

## 1 Introduction

Summarizing the response of the Earth system to anthropogenic forcers with metrics has long been practised as a way to illustrate uncertainty in Earth system response to greenhouse gases. For example, the concept of the equilibrium climate sensitivity (ECS), the equilibrium global mean temperature increase which would be observed in response to a doubling of atmospheric carbon dioxide concentrations (Hansen et al., 1984), has existed for over 50 years (Charney et al., 1979) and a significant amount of literature has been devoted to constraining its value (Knutti et al., 2017).

The Earth system responds to a step change in forcing on timescales ranging from days to millennia (Knutti and Rugenstein, 2015), so an "effective climate sensitivity" (EffCS here on) is often used as a proxy for decadal to centennial feedbacks. EffCS is generally calculated in a coupled atmosphere–ocean model from the output of the

"abrupt4xCO2" simulation, a standard experiment in which $CO_2$ concentrations are quadrupled instantaneously from pre-industrial levels and the model is allowed to evolve (Gregory et al., 2004).

EffCS is calculated by assuming that a model is associated with a single feedback parameter (i.e. a rate of change of top of atmosphere radiative flux per unit surface temperature increase), allowing the equilibrium temperature response to a step change forcing to be predicted by linear extrapolation. Another metric, the transient climate response (TCR) at the time of $CO_2$ doubling or quadrupling (T140) is calculated from an "1pctCO2" idealized experiment in which $CO_2$ concentrations are increased by 1 % each year, starting from a pre-industrial state, resulting in linearly increasing forcing.

Although it was generally assumed that TCR would be a better predictor of transient warming under a high emissions scenario such as Representative Concentration Pathway 8.5

(RCP8.5) (Riahi et al., 2011), a complication has arisen due to the fact that EffCS seems to be better correlated than TCR with 21st century warming from present-day levels under a business-as-usual scenario (Grose et al., 2018). The reason for this is not yet well understood given that the radiative pathway in RCP8.5 leading up to 2100 is relatively similar to that of the 1 % annual increase experiment used to measure T140. Furthermore, neither EffCS nor TCR is well correlated with end-of-century temperatures in a mitigation scenario (Grose et al., 2018) such as RCP2.6 (van Vuuren et al., 2011), which calls into question the relevance of such summary metrics in the discussion of mitigation adaptations.

Similarly, a number of studies have shown that the EffCS approximation does not well describe the true equilibrium behaviour of most models (Knutti et al., 2017). When general circulation model (GCM) abrupt4xCO2 simulations are continued for thousands of years, many are found to deviate significantly from the linear trend line one would fit to a 150-year simulation (Andrews et al., 2015; Knutti et al., 2017; Senior and Mitchell, 2000; Rugenstein et al., 2016).

The conceptual models representing the evolving feedbacks as a function of timescales vary slightly between studies – either modulating the efficacy of deep ocean heat uptake (Geoffroy et al., 2013; Winton et al., 2010; Held et al., 2010) or by representing the climate system as sum of warming patterns which emerge on different adjustment timescales (Armour et al., 2013; Rugenstein et al., 2016), each associated with their own feedback parameter. However, the analytical set of solutions for the temperature response to a step change in forcing is the same in either case – a superposition of decaying exponential modes with different timescales varying between a few years and a few centuries (Proistosescu and Huybers, 2017). It has been shown that the implications of these additional degrees of freedom and ambiguity over contributions from different timescales of response might imply that EffCS may not be strongly constrained by temperature change over the last century (Proistosescu and Huybers, 2017; Andrews et al., 2018), and that the long-term equilibrium (LTE) sensitivity may be greater than that implied by EffCS (Otto et al., 2013; Lewis, 2013).

This state of understanding leads to a number of emerging critical questions which we discuss in this paper – can we explain the non-intuitive result that EffCS is a better predictor than T140 of end-of-century temperatures under RCP8.5? Which summary metrics of global sensitivity to greenhouse gas forcing are most useful for effective policy decisions? Finally, do the implicit structural assumptions underpinning the applicability of these metrics to the real world cause us to mis-categorize and potentially underestimate future warming risk?

## 2   A simple model example

We begin by considering an idealized ensemble of climate model simulations. We use a two-timescale thermal response model, conceptually representing the deep ocean (with a response timescale of a century or more) and shallow ocean response timescales (with a response timescale of 10 to 50 years). Such a model, although simple, is capable of resolving evolving feedback amplitudes and can emulate the climatological responses of complex Earth system models on two timescales. Such a model makes a structural assumption that the Earth can be modelled as a discrete sum of linear decaying exponential responses to forcing, but this model has been found to well describe GCM evolution on a century timescale (Proistosescu and Huybers, 2017; Geoffroy et al., 2013) and is sufficiently complex to illustrate the limitations of defining system sensitivity through TCR or EffCS.

To efficiently describe the response of the system to a generic forcing, this study employs a linear Green function which describes the forcing by convolution with an impulse response (Ruelle, 1998 – in this case, the step change in $CO_2$ forcing). This approach can be used to approximate and simplify global climate dynamics (Ragone et al., 2015; Lucarini et al., 2017), and its computational efficiency allows Markov chain Monte Carlo parameter estimation for the physical parameters. Furthermore (and critically for this study), the pulse–response formulation can be used to self-consistently relate different metrics of climate sensitivity on a range of timescales (Lucarini et al., 2017).

### 2.1   Model formulation

The two-timescale impulse response model follows the thermal feedback-timescale implementation from the FAIR simple climate model (Smith et al., 2018; Millar et al., 2017), which follows Hasselmann et al. (1993):

$$\frac{dT_n}{dt} = \frac{q_n F - T_n}{d_n}; \; T = \sum_n T_n; \; n = 1, 2, \tag{1}$$

where $T_n$ is global mean temperature and for each timescale $n$. $T_n$ is the component of warming associated with that timescale, $q_n$ is the feedback parameter, and $d_n$ is the response timescale.

Note the use of $n = 2$ timescales is a structural choice, used here both for relevance to parameterization choices in commonly used simple models (Smith et al., 2018; Geoffroy et al., 2013; Goodwin et al., 2018; Meinshausen et al., 2011) and because the parameters of two-timescale model can be readily interpreted and unambiguously fitted to complex model output. The $n = 1$ timescale provides a significantly poorer fit to temperature evolution in abrupt4xCO2 Coupled Model Intercomparison Project (CMIP) simulations (see Fig. S5). Notably, some authors have considered three timescale models (Caldeira and Myhrvold, 2013; Joos

https://doi.org/10.5194/esd-11-1-2020

et al., 2013; Tsutsui, 2017) or general linear response functions (Ragone et al., 2015; Lucarini et al., 2017; Lembo et al., 2020) which allow (effectively) for an unlimited number of exponential response modes (Lucarini, 2018). While we observe a small further improvement in fit is apparent for some models with $n = 3$ modes, not all models appear to express three response timescales, which causes unstable fitting behaviour in those cases and a difficulty in comparing and interpretation of the values of fitted parameters across CMIP. Nevertheless, further understanding the feedback timescale dynamics of different CMIP models is an important topic for further research.

Total heat flux into the system $R$ is divided into shallow and deep ocean fluxes, defined as a function of the same two timescales:

$$R = \sum_{n=1}^{2} R_n; \tag{2}$$

$$\text{s.t.} \sum_{n=1}^{2} r_n = 1; \quad R_n = r_n (F - T_n/q_n); \tag{3}$$

where $r_n$ is an efficacy factor for heat absorbed by the deep ($n = 1$) or shallow ($n = 2$) ocean, which sum to unity given the boundary condition that $R(0) = F(0) = F_{4xCO2}$ at $t = 0$ (allowing just one degree of freedom $r_1$ – the fraction of heat which is allocated to deep ocean storage).

The particular solutions for temperature and radiation response to a step change in forcing $F_{4xCO2}$ at time $t = 0$ can be expressed as a sum of exponential decay functions:

$$T_p(t) = F_{4xCO2} \sum_{n=1}^{2} q_n (1 - \exp(-t/d_n)) \tag{4}$$

$$R_p(t) = F_{4xCO2} \sum_{n=1}^{2} r_n (\exp(-t/d_n)), \tag{5}$$

where $T_p(t)$ is the annual global mean temperature, $R_p(t)$ is the net top-of-atmosphere (TOA) radiative imbalance at time $t$, and $F_{4xCO2}$ is the instantaneous global mean radiative forcing associated with a quadrupling of $CO_2$, taken here to be $7.4\,W\,m^{-2}$ (Myhre et al., 2013).

We define a historical forcing time series as a function of $CO_2$ concentrations $C(t)$ and a non-$CO_2$ forcing time series $F_{nonCO2}(t)$ (both taken from Meinshausen et al., 2011):

$$F(t) = \frac{F_{4xCO2}}{\ln(4)} \ln\left(\frac{C(t)}{C_0}\right) + f_r F_{aer} + F_{other}, \tag{6}$$

where $f_r$ is a free parameter to allow scaling of aerosol forcing (conceptually allowing for forcing uncertainty in the historical time series), and $F_{other}$ is all other anthropogenic and natural forcers (summed from Meinshausen et al., 2011). The thermal response is calculated by expressing the numerical time derivative of the forcing time series $F(t)$ where the change in forcing in a given time step in a given year $\Delta F(t')$

is $[F(t') - F(t' - 1)]$. The forcing time series can thus be expressed a series of step functions, and $T_p$ from Eq. (4) can be used to calculate the integrated thermal response.

$$T(t) = \sum_{t'=0}^{t} \Delta F(t') \sum_{n=1}^{2} q_n \left(1 - \exp\left(\frac{-(t - t')}{d_n}\right)\right) \tag{7}$$

Heat fluxes into the deep ($D(t)$) and shallow ($H(t)$) ocean components are represented by numerical integration of the slow ($n = 1$) and fast ($n = 2$) pulse response components of $R_p(t)$ in Eq. (5):

$$D(t) = r_1 \sum_{t'=0}^{t} \Delta F(t') \exp\left(\frac{-(t - t')}{d_1}\right), \tag{8}$$

$$H(t) = (1 - r_1) \sum_{t'=0}^{t} \Delta F(t') \exp\left(\frac{-(t - t')}{d_2}\right). \tag{9}$$

### 2.1.1 Model optimization

The model input time series for calibration are observed $CO_2$ concentrations, along with radiative estimates from Meinshausen et al. (2011) of non-$CO_2$ forcing agents. We optimize the thermal model parameters for two timescales and the non-$CO_2$ forcing factor (see Table 1).

A Markov chain Monte Carlo (MCMC) optimization procedure produces an ensemble of parameter configurations such that the density of the simulations in parameter space reflects the likelihood as reflected in a cost function (as represented by a number of pre-defined likelihood metrics). MCMC algorithms employ a random walk in parameter space which ultimately seeks to produce a representative sample of the distribution.

The classical approach to this random walk is the Metropolis–Hastings algorithm (MacKay, 2002), which iteratively moves a set of "walkers" or sample points throughout the parameter space. This approach, however, is computationally inefficient, as it requires the specification of the transition distribution with a large number of degrees of freedom. Here, we follow the Goodman and Weare (2010) MCMC implementation which updates a walker position using a vector defined stochastically from the remaining ensemble of walkers. This approach has fewer degrees of freedom and is a well-tested approach for multidimensional optimization problems (Foreman-Mackey et al., 2013). We use flat initial parameter distributions as shown in Table 1, 200 walkers and 50 000 iterations for each optimization.

**Table 1.** A table showing model parameter values and minimum and maximum values allowed in model optimization.

| Long name | Symbol | Min | Max |
|---|---|---|---|
| Thermal adjustment of deep ocean sensitivity ($KW\,m^{-2}$) | $q_1$ | 0 | 10 |
| Thermal adjustment of upper ocean sensitivity ($KW\,m^{-2}$) | $q_2$ | 0 | 10 |
| Thermal adjustment of deep ocean timescale (years) | $d_1$ | 100 | 4000 |
| Thermal adjustment of upper ocean timescale (years) | $d_2$ | 10 | 100 |
| Fraction of forcing in deep ocean response | $r_1$ | 0.0 | 1 |
| Non-$CO_2$ forcing ratio | $f_r$ | 0.7 | 1.3 |

Cost functions are computed for global mean temperature, shallow and deep ocean content:

$$E_T = \sum_t \left( \frac{(T(t) - T_{obs}(t))}{\sqrt{2}\sigma_T} \right)^2 \qquad (10)$$

$$E_H = \sum_t \left( \frac{(H(t) - H_{obs}(t))}{\sqrt{2}\sigma_H} \right)^2, \qquad (11)$$

$$E_D = \sum_t \left( \frac{(D(t) - D_{obs}(t))}{\sqrt{2}\sigma_D} \right)^2, \qquad (12)$$

where $T_{obs}$ is HadCRUT 4.6 ensemble median global mean temperature anomalies (Morice et al., 2012) relative to a 1850–1900 baseline, and $\sigma_T$ is defined as the standard deviation of HadCRUT 1850–1900 values. Shallow and deep ocean heat fluxes are taken as the 0–300 and 300 m plus heat content derivatives, respectively, in Zanna et al. (2019), with $\sigma_H$ and $\sigma_D$ taken as 1850–1900 standard deviations from the same dataset.

Flat priors are used for all parameters, with an additional prior on true equilibrium climate sensitivity using the likely value and upper bound on equilibrium climate sensitivity from Goodman and Weare (2010) to specify the median and 90th percentile of a gamma distribution for equilibrium sensitivity (i.e. warming as $t \to \infty$).

We demonstrate that this technique is able to capture the broad uncertainty associated with future projections of CMIP models by using pre-2020 temperatures in RCP8.5 to calibrate the simple model outlined above (see Fig. S3). In most cases, the future projection for each scenario falls within the distribution arising from the MCMC ensemble fit, with some specific exceptions – FIO-ESM, FGoals-G2, CCSM4 (which share some common heritage) and the NASA Goddard Institute for Space Studies (GISS) models. As such, the observationally fitted MCMC ensemble explores broadly comparable uncertainty to that seen in the bulk of the CMIP ensemble, with the caveat that the ensemble tends to undersample cases where there is little or no long-term warming response to emissions.

The physical parameters of this simple model are constrained by historical carbon dioxide concentrations together with observed global mean temperatures from 1870 to the present day (together with aggregate forcing estimates representing other anthropogenic emissions (Meinshausen et al.,

2011), which are not the focus of this study). The posterior parameter distribution for the model can then be used to project the corresponding range of response in probabilistic projections of the future scenarios or in idealized experiments which simulate a range of self-consistent values for various climate sensitivity metrics.

### 2.1.2 Idealized simulations

Effective climate sensitivity is measured by implementing a step change abrupt $CO_2$ quadrupling and following Gregory et al. (2004) to assess the linear extrapolation of warming at the point of net TOA energetic balance. A140 is calculated as the average of years 131–150 of the abrupt4xCO2 simulation. TCR and T140 are calculated as the average of years 61–80 and 131–150, respectively, of the 1pctCO2 simulation (during which the $CO_2$ concentrations are doubled and quadrupled, respectively), where $CO_2$ concentrations are increased annually by 1 %, resulting in a linear increase in climate forcing. RCP scenario temperature trajectories are calculated for each parameter set using concentration and forcing time series from Meinshausen et al. (2011) from 1850 to 2300.

Resulting EffCS values (to a doubling of $CO_2$) range from 2.4 to 4.6 K (5th and 95th percentiles) and values of TCR from 1.6 to 2.2 K (Fig. 1b and e). This results in a range of 21st century warming under two scenarios considered: RCP2.6 (RCP8.5) 2100 warming ranges from 1.4 to 2.4 K (3.8 to 5.1 K), respectively (5th and 95th percentiles; see Fig. 1a).

We then consider, in the context of this observationally constrained ensemble of simple models, which idealized metrics of system response are most informative for describing 21st century warming. We consider a number of sensitivity metrics: the EffCS, TCR and T140 (transient warming under an annual compounded 1 % increase in $CO_2$ concentrations at the time of $CO_2$ doubling and quadrupling, corresponding to years 70 and 140 of the simulation). Finally, we consider A140 as a possible metric for consideration, defined as the global mean warming above pre-emission levels in the abrupt4xCO2 simulation calculated 140 years after time of $CO_2$ quadrupling (here and throughout estimated as the mean from years 131 to 150). Figure 2 illustrates how ensemble

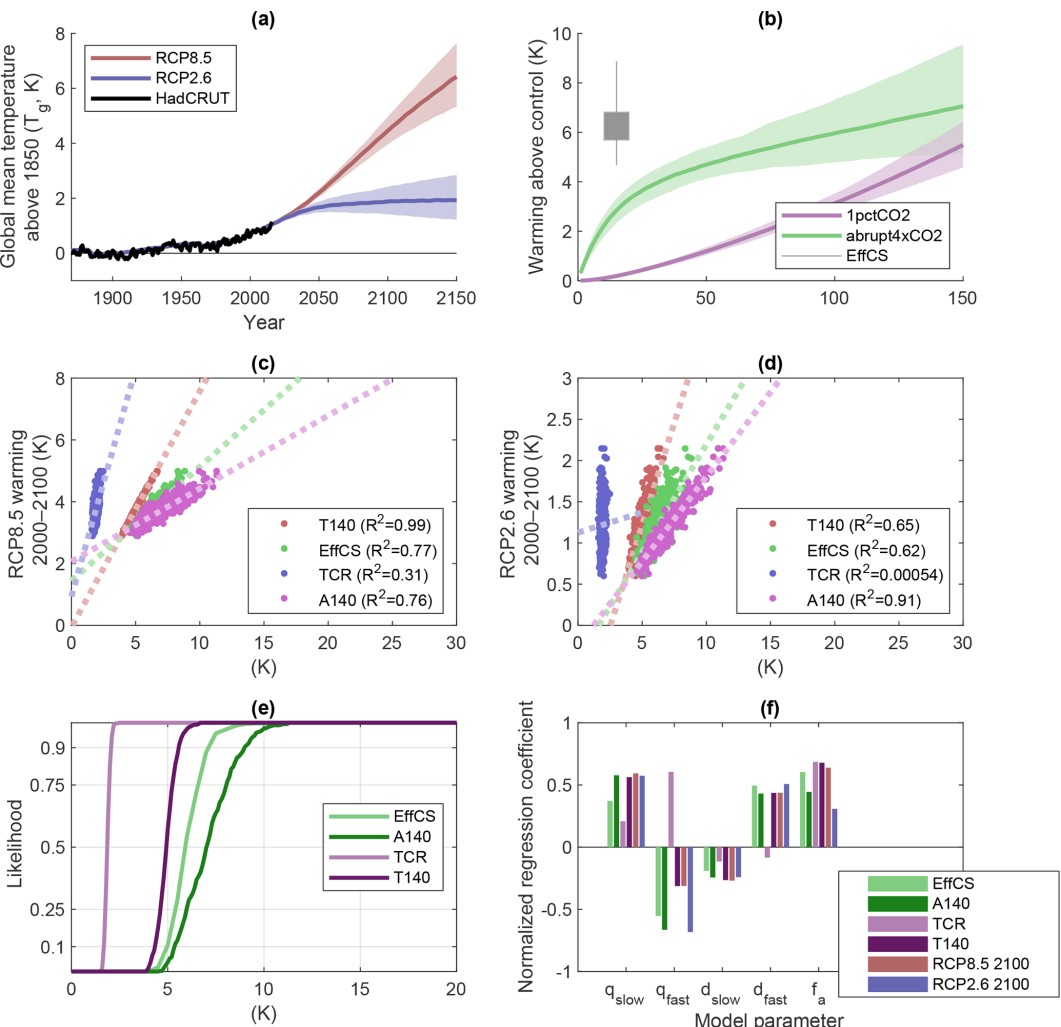

**Figure 1.** An observationally constrained ensemble of simple models. Panel **(a)** shows the global mean temperature both historically and under the RCP2.6 and RCP8.5 scenarios. Black lines show the HadCRUT data used in calibration, whereas shaded regions show the 10 %–90 % range of scenario projections in the posterior simple model ensemble distribution. Panel **(b)** shows the corresponding time series posterior distributions for the abrupt4xCO2 and 1pctCO2 simulated experiments, with grey error bars showing range of EffCS for $CO_2$ quadrupling (boxes and whiskers show 25th–75th and 1st–99th percentiles, respectively). Panels **(c, d)** show relationships between different sensitivity indicators and 2000–2100 temperature changes under RCP8.5/RCP2.6, respectively; panel **(e)** shows the posterior cumulative probability density functions for the four sensitivity variables considered; and panel **(f)** shows the parameter regression coefficients relating the five normalized model input parameters to the four normalized sensitivity metrics.

spread would be impacted for a set of different scenarios if each of these metrics were constrained to lie within a narrow range (nominally the 45–55th percentile range of values present in the entire observationally constrained ensemble).

In the high emissions RCP8.5 scenario (Riahi et al., 2011), 2000–2100 warming is nearly perfectly described ($R^2 = 0.99$) by T140, the transient climate response after 140 years in a 1 % $CO_2$ simulation (Figs. 1c and 2k). The corresponding response after only 70 years (TCR) is a much poorer predictor at $R^2 = 0.31$.

These results are physically intuitive. The climate forcing and rate of change of forcing in RCP8.5 at the end of the

21st century are of similar magnitude to those in year 140 of the 1 % $CO_2$ simulation, and so it is unsurprising that T140 is an efficient predictor for RCP8.5. TCR is a poor predictor in the simple model ensemble largely because TCR itself is already highly constrained by historical warming (Fig. 1e), and thus the ensemble is effectively conditioned on a value of TCR and it has little additional explanatory value in explaining the ensemble variance in the RCP projections (Fig. 2f and g).

EffCS and A140 are also well correlated with the RCP8.5 warming ($R^2 = 0.77$ and 0.76, respectively) but less so than T140. For the mitigation scenario (RCP2.6), the most effec-

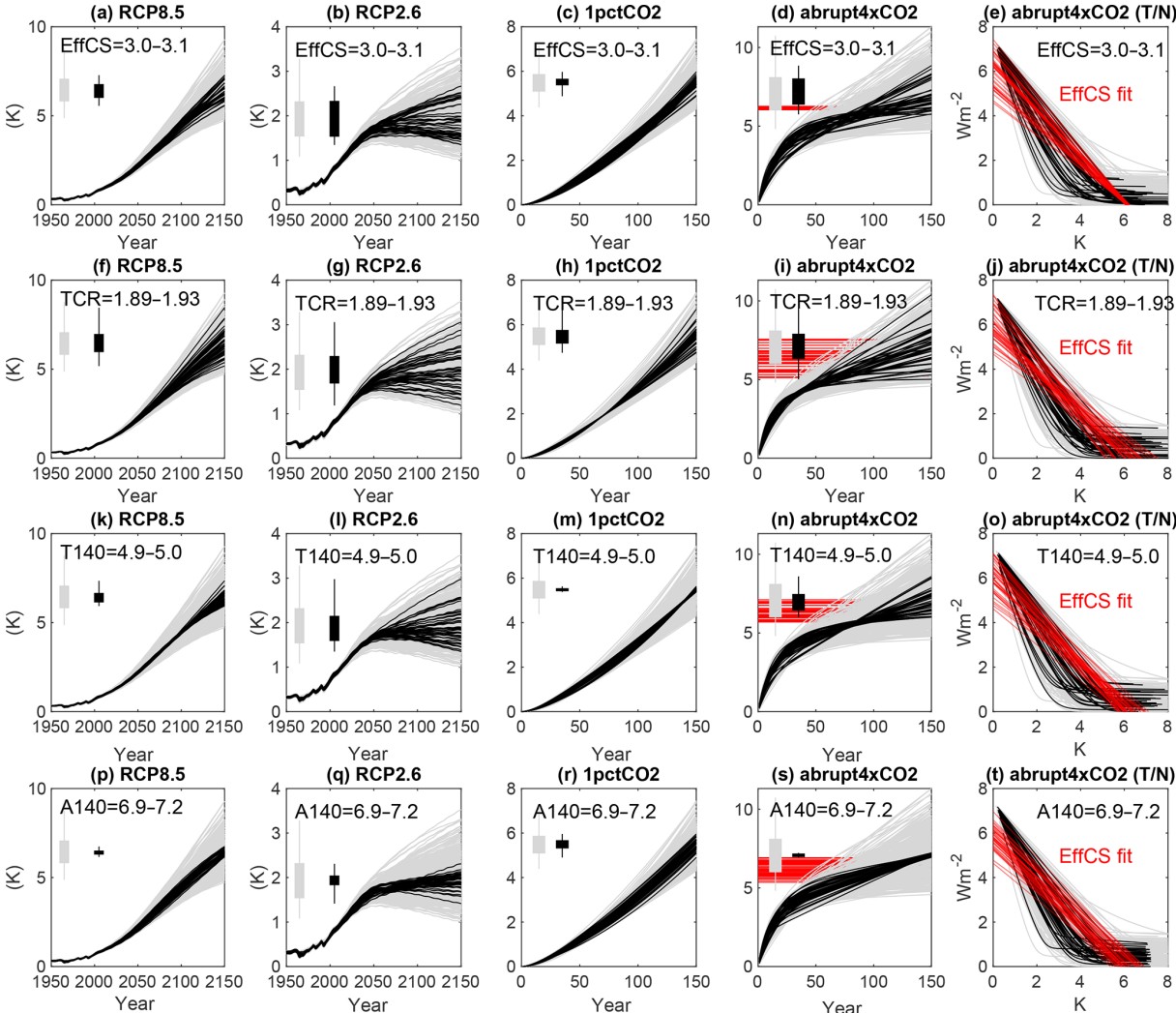

**Figure 2.** An illustration of how constraining different types of global sensitivity metric impact the idealized spread of global mean temperature evolution under different scenarios. Each row illustrates one constraint: effective climate sensitivity to $CO_2$ doubling (EffCS), TCR (70 years, $CO_2$ doubling), T140 (140 years, $CO_2$ quadrupling) and A140. Lines in grey show the entire posterior distribution of models from Fig. 1, while lines in black show the 45th–55th percentiles of the distribution of the respective quantity. Panels **(a–s)** show global mean temperature time series of a scenario or idealized experiment – RCP8.5, RCP2.6, 1 % ramping $CO_2$, abrupt CO2 quadrupling (the fifth column shows energetic imbalance as a function of surface temperature in the abrupt4xCO2 experiment). Histograms show the resulting distribution of temperature in 2150 (RCP8.5/2.6) or year 140 (1pctCO2, abrupt4xCO2) for the complete distribution (grey) and 45th–55th percentile range (black). Red lines show the distribution of values of effective climate sensitivity **(d, i, n, s)** and the trend lines used to compute it **(e, j, o, t)**.

tive predictor of 2000–2100 warming is A140 ($R^2 = 0.91$). Both EffCS and T140 are weakly correlated ($R^2 = 0.62$ and 0.65, respectively), and TCR shows no significant correlation.

To help understand these relationships, we can perform a regression analysis of the metrics as a function of model ensemble parameters (Fig. 1f), which suggests A140 and RCP2.6 warming from 2000 to 2100 is controlled by the *difference* between the slow and fast components of sensitivity. We can understand this in the context of the way the model is constrained by historical temperatures.

There is a trade-off between fast and slow components of climate sensitivity in the posterior parameter distribution of the ensemble (see Fig. 3), which broadly determines the fraction of equilibrium warming associated with current forcing levels that has already been experienced. There is also a correlation between fast sensitivity and fast timescale. These relationships should be broadly expected if we consider that the observed transient warming of the model has been constrained by the model. If we consider the analytical expression for TCR (warming after 70 years of 1 % annual increase in $CO_2$) in a two-timescale model (from Eq. 7 following

Earth Syst. Dynam., 11, 1–15, 2020                                    https://doi.org/10.5194/esd-11-1-2020

Smith et al., 2018):

$$\text{TCR} = F_{2\text{xCO2}} \Big[ \Big( q_1 \Big( 1 - d_1/70 \Big( 1 - e^{-70/d_1} \Big) \Big) \Big)$$
$$+ q_2 \Big( 1 - d_2/70 \Big( 1 + e^{-70/d_2} \Big) \Big) \Big], \tag{13}$$

where $F_{2\text{xCO2}}$ is the forcing from a doubling of atmospheric $CO_2$, $q_1$, $d_1$ are the fast sensitivity and timescale, and $q_2$, $d_2$ are the slow sensitivity and timescale. In the limit that $d_1 \ll 70$ and $d_2 \gg 70$, we obtain the following:

$$\text{TCR} \approx F_{2\text{xCO2}} \left( \frac{q_1}{1 + \frac{d_1}{70}} + \frac{q_2}{2} \left( \frac{70}{d_2} \right)^2 \right). \tag{14}$$

This expression explains the primary features apparent in the MCMC posterior distribution if we consider that the observations broadly fix the value of TCR: an inverse relationship is expected between $q_1$ and $q_2$, and we observe this in Fig. 3. The fast component (left-hand term in Eq. 14) is constrained by the historical warming time series to be non-zero (see Fig. 3) – and there is a tight proportionality in constrained values of $q_1$ and $d_1$. Only the lower bound of the slow timescale $d_2$ is constrained for a given value of $q_2$; i.e. the transient warming alone provides no information on the upper bound of the slow response timescale.

Thus, if a greater fraction of today's observed warming is explained with the faster component of model response, we would expect less unrealized warming in a mitigation scenario later in the century. This causes large uncertainties in RCP2.6 evolution in the constrained ensemble, even in the case that we had confidence in the values of EffCS, TCR or T140 (Fig. 2b, g and l).

The constrained distribution for fast-timescale sensitivity is near Gaussian and non-zero in all ensemble members, whereas slow-timescale sensitivity is more weakly constrained by the observations ranging from near-zero to large ($20 \, \text{K W m}^2$) long-term equilibrium responses. The slow feedback component strongly controls A140 and RCP2.6 warming (Figs. 1d, f and 2q).

RCP8.5 warming and T140, however, are associated with a near-linear increase in forcing throughout the simulation which results in a near-linear temperature increase. The relative fraction of warming associated with fast- and slow-timescale feedbacks remains constant over time, and thus warming to date (effectively fixing TCR, subject to aerosol forcing uncertainty) better constrains relative error in future response in a non-mitigation scenario (Fig. 2f).

## 3 Considering the multi-model ensemble

But how do the findings in the simple model framework reconcile with findings in the CMIP5 and CMIP6 multi-model ensembles? Firstly, it is plausible that there is some commonality in the lack of skill of TCR (the transient response after 70 years) in our simple model ensemble and in the CMIP ensembles. In our simple model case, the ensemble members were explicitly calibrated to reproduce the 20th and early 21st century warming – which is a very strong constraint on the value of TCR in this idealized setup.

Earth system model calibration is conducted in a much larger parameter space by groups with a wide range of objectives which complicate interpretation (Mauritsen et al., 2012; Sanderson and Knutti, 2012), but simulations are generally only published using models which are able to adequately describe the 20th century and thus might be subject to a similar effective constraint on TCR which renders the metric ineffective for describing variance in the future evolution of the model. But there remains a direct contradiction for T140, where the simple model suggests T140 should be a better predictor than EffCS for non-mitigation warming in the 21st century, whereas the opposite was found in the CMIP correlations (see Fig. S2 and Grose et al., 2018).

To understand this, we need to consider how the properties of the simple model ensemble differ from the CMIP archive. Although the thermal response of the simple model is broadly able to represent the climatological response of CMIP models to step forcing and transient forcing in $CO_2$ over a century timescale (Geoffroy et al., 2013; Proistosescu and Huybers, 2017), it contains no internal climate variability, and all experiments in Sect. 2 are conducted from an idealized, perfectly spun-up state.

Both of these assumptions are not true for CMIP5 or CMIP6. Measurements of EffCS and TCR are complicated by internal variability (Knutti and Rugenstein, 2015), and many models still exhibit some temperature drift in the control simulation from which the 1pctCO2 simulations and abrupt4xCO2 simulations are branched (Fig. 4). This creates uncertainty from two sources – firstly, it is not always apparent at what point during the control simulations the 1pctCO2 simulation has been branched; thus, there is uncertainty in how the anomaly should be measured. Secondly, there is the potential for an unknown contribution of control drift to be erroneously included in the temperature evolution of the 1pctCO2 and abrupt4xCO2 simulations.

To assess the contribution of control drift bias in sensitivity metrics, we implement idealized representations of non-equilibration into our simple model from Sect. 2. We then create an idealized distribution of drift similar to that seen in the CMIP ensembles in the simple model ensemble by initializing the model 500 years before the experiment begins, defining an effective "baseline" period from which anomalies are measured to be the average temperature between the years 400 and 500. Climate internal variability is represented by a second-order autoregressive model, which is fitted to each CMIP model in turn. The ensemble-mean autoregressive parameters are used to create artificial "noisy" simulations by linearly adding noise generated from the autoregressive model to the output of the simple model.

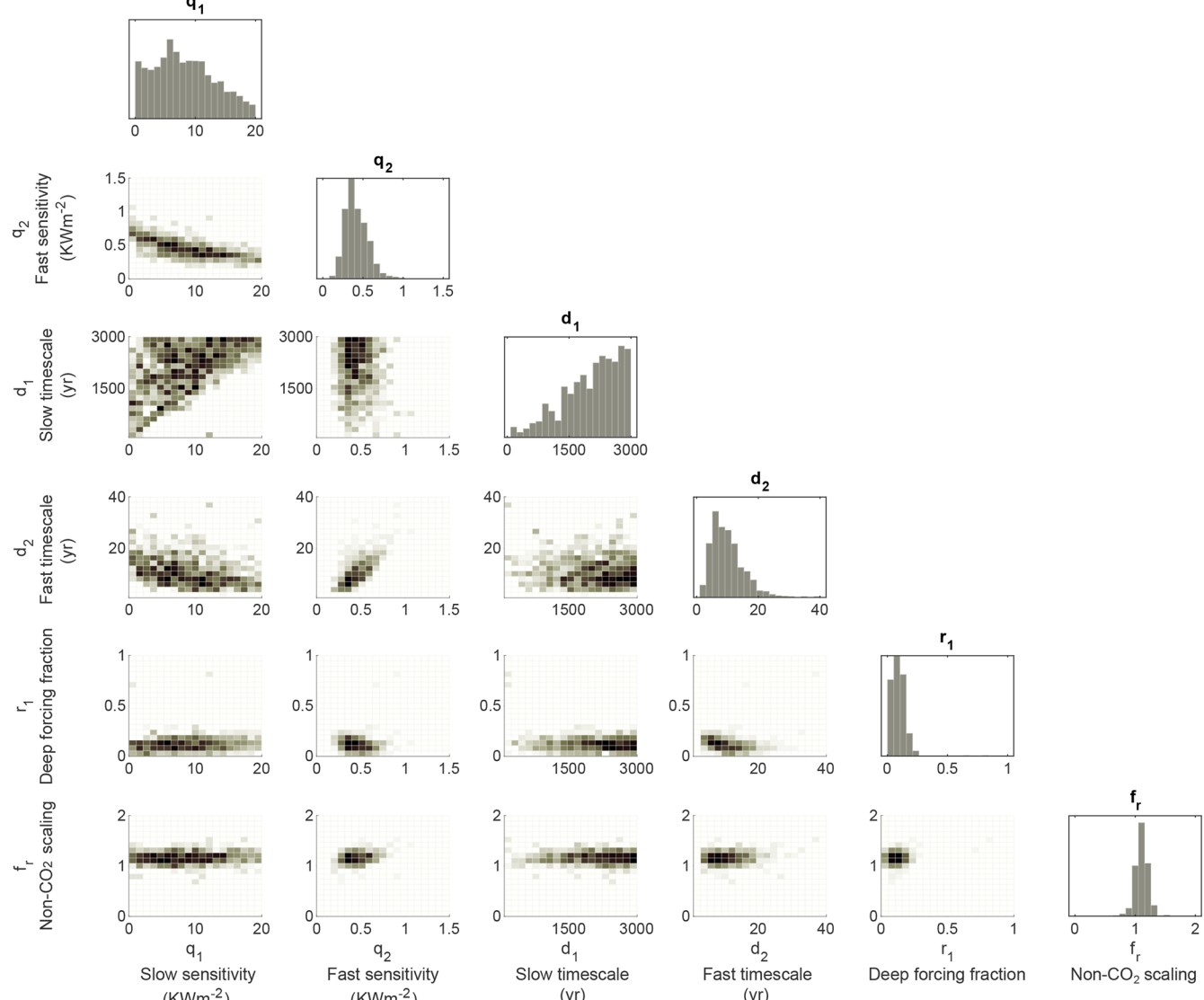

**Figure 3.** A "corner plot" showing the posterior parameter distribution attained by MCMC calibration of the simple climate model. Diagonal plots show posterior histograms for each of the parameter values optimized in the calibration, where the $x$-axis range reflects the bounding values of the initial flat prior distribution. Off-diagonal plots show pairwise distributions of parameters in the posterior distribution.

We consider the range of control drifts observed in the CMIP5 and CMIP6 ensembles (illustrated in Fig. 4l) which range from $-0.3$ to $+0.6$ K per century in the CMIP5 and CMIP6 models considered in this study. An idealized distribution of drift in the simple model ensemble is created by initializing the model 500 years before the abrupt4xCO2 or 1pctCO2 simulation with a non-zero, constant forcing drawn from a flat distribution ranging from $-1$ to $+1$ W $m^{-2}$, which results in a distribution of control drift of $-0.4$ to $+0.4$ K per century (i.e. broadly comparable to the CMIP case). For each simulation, we consider a baseline for temperature to be defined by the average global mean temperature in years 400–500.

To represent the first-order effect of climate noise, we fit a second-order autoregressive model to the detrended global mean temperature time series in each available model in the CMIP5/6 ensemble. Taking CMIP mean parameters for the variance and autoregressive parameters, we generate noise for each realization of the simple model (though we note, in practice, that the noise characteristics vary by CMIP model).

The results are illustrated in Fig. 5a, where the simple model ensemble is initialized in a non-equilibrium state with additive Gaussian noise. With these additional sources of error, both EffCS and A140 are not strongly impacted when measured in the noisy/non-equilibrated model variants (Fig. 5b and c), but the T140 measurement is strongly degraded (Fig. refdriftd). Indeed, in this ensemble, the biased

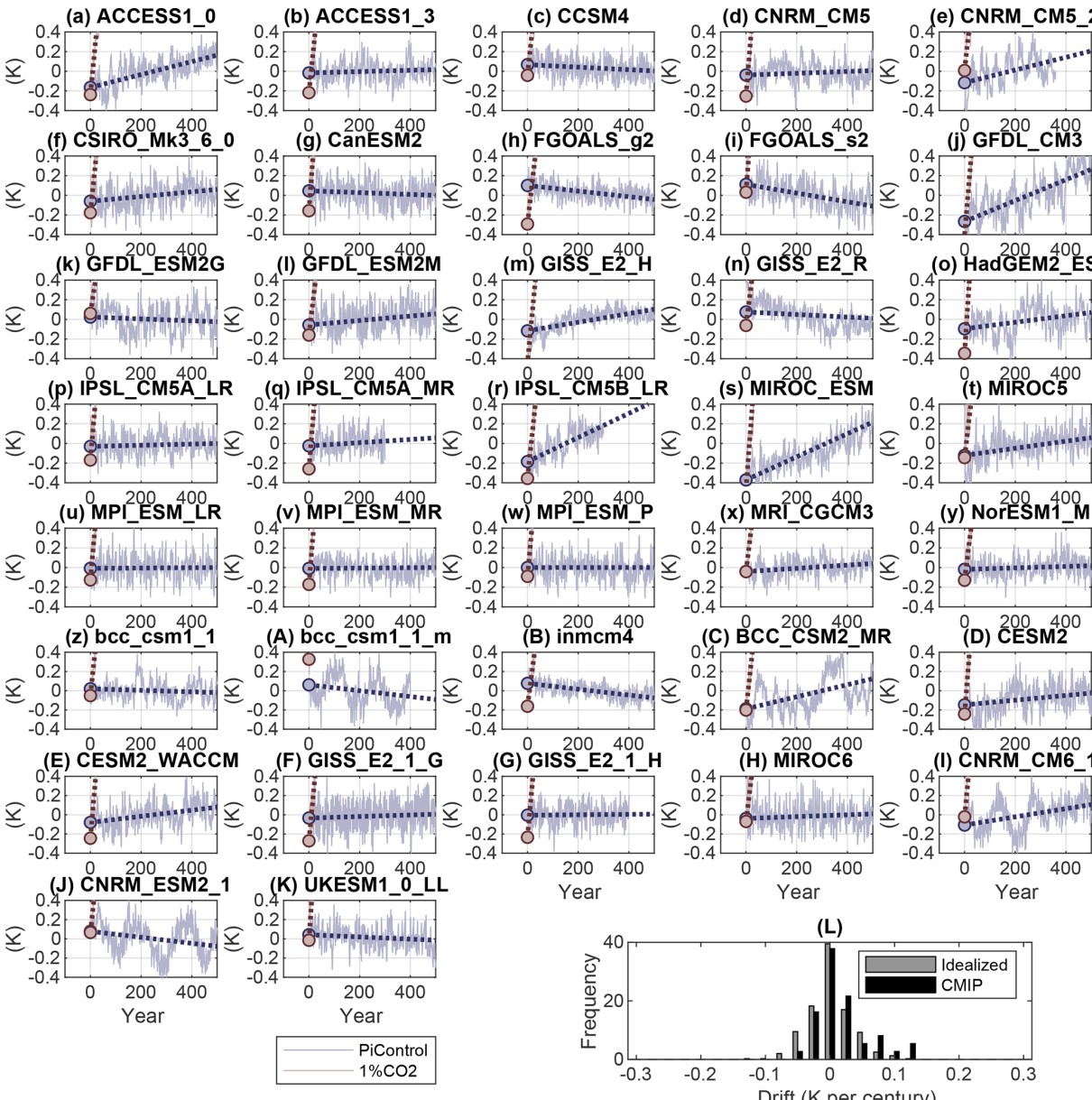

**Figure 4. (a–k)** Control simulation global mean temperatures from a selection of models in the CMIP5 and CMIP6 ensembles. Control simulations (blue) and initial years of 1pctCO2 simulations (pink) are plotted. Dotted lines show linear fit to the available time series. Blue and pink circles show the intersection of the linear temperature fit at the start of the simulation. **(l)** Histogram showing the distribution of control model trend in CMIP (black) and in an idealized ensemble of non-equilibrated simple models considered in Fig. 5 (grey).

measurements of EffCS or A140 are slightly better correlated with true T140 than the biased measurement of T140 itself. This provides a possible explanation for why T140 may be a poor predictor of RCP8.5 warming in CMIP.

In our simple framework, the reasons for the more accurate measurement of EffCS are primarily associated with the lack of equilibration. Simply adding noise from the autoregressive model has little effect on the accuracy of EffCS, T140 or A140 (where both T140 and A140 are estimated using the average of years 131 to 150 in the simulation; see Table 2).

Both A140 and EffCS are less sensitive to non-equilibrated initial states than T140. The former experiences the same variance due to the uncertain climate drift, but the absolute value of A140 tends to be larger than T140; thus, there is less relative error in its estimation. The effect on the drift on EffCS is muted because the near-linear climate drift primarily biases the estimation of slow rather than fast feedbacks (see Fig. S1 in the Supplement). Because EffCS is primarily a measure of fast-mode feedback strength (see Fig. 1f),

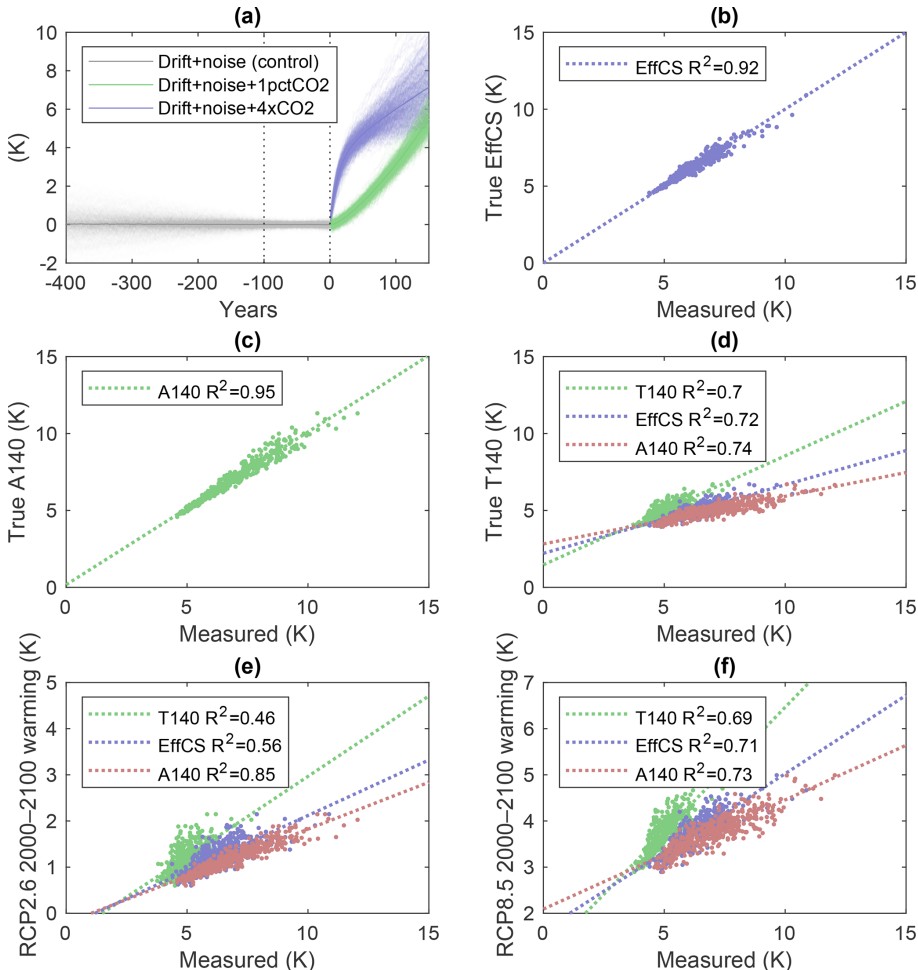

**Figure 5.** An idealized ensemble of simple models, where model parameters are identical to those considered in Fig. 1b, but models are initialized in a non-equilibrium state such that the baseline period is subject to some control drift, and model output is also subject to interannual variability of a similar magnitude to models in the CMIP archive. Panel **(a)** shows global mean temperature evolution for the control period (grey), abrupt4xCO2 simulation (blue) and 1pctCO2 simulation (green). Panels **(b, c)** show the true value of (EffCS, A140) as calculated in the noise-free, equilibrated simulations, plotted as a function of the measured value of (EffCS, A140) in a noisy, non-equilibrated simulation. Panels **(d, f, g)** show the true value of (T140, RCP2.6, RCP8.5 2000–2100 warming) plotted as a function of the measured values of T140, EffCS and A140, respectively.

its value is less impacted if experiments are started from a non-equilibrium state.

There is some evidence that the lack of equilibration has an outsized effect on the estimation of TCR in the CMIP models. In Fig. 6, we attempt to unbias the estimate of TCR in two ways. Firstly, we estimate the baseline temperature by regressing the temperatures in the first 20 years of the 1 % $CO_2$ ramp experiment as a function of time (see Fig. S4). Anomalies in temperature (and TOA fluxes for ECS) are measured relative to the corrected baselines derived from the 1pctCO2 simulation, and estimated linear pre-industrial trends are subtracted from the 1pctCO2 and abrupt4xCO2 time series. This pre-processing of the temperature time series improves the correlation between TCR and 21st century warming under RCP8.5 from 0.86 to 0.89. It also improves the correlation

between EffCS and 21st century warming slightly from 0.94 to 0.95 (and A140 from 0.89 to 0.91).

These "corrected" values (listed in Table 3) are estimates only, given that we would expect the regression estimate based on a short 20-year period to be itself subject to internal variability noise, and we are assuming that the abrupt4xCO2 simulation and 1pctCO2 simulation have the same baselines. However, the improvement in correlation with future warming seen over the case with the pre-industrial average baseline supports the hypothesis that control drift adds uncertainty to the estimation of all quantities (and particularly TCR). However, it is not a complete explanation –and even after this adjustment, EffCS remains better correlated to RCP8.5 transient warming than TCR in the multi-model ensemble.

**Table 2.** A table showing $R^2$ regression statistics relating a set of predictors to a set of unbiased model properties. Predictors are transient climate sensitivity at quadrupling of $CO_2$ (T140), effective climate sensitivity (EffCS) and warming 140 years after a quadrupling of $CO_2$ (A140); additional rows show these values measured experiments conducted with non-equilibrated base climates (drift), additive autoregressive noise (noise) and a combination of both factors (drift plus noise). "True" output model properties (T140, EffCS, A140, RCP8.5 and RCP2.6 warming from 2000 to 2100) are derived from the equilibrated model without noise.

| Predictor | T140 | EffCS | A140 | RCP8.5 2000–2100 | RCP2.6 2000–2100 |
|---|---|---|---|---|---|
| T140 (true) | 1.00 | 0.78 | 0.77 | 0.99 | 0.65 |
| EffCS (true) | 0.78 | 1.00 | 0.70 | 0.77 | 0.62 |
| A140 (true) | 0.77 | 0.70 | 1.00 | 0.76 | 0.91 |
| T140 (drift) | 0.74 | 0.58 | 0.59 | 0.73 | 0.50 |
| EffCS (drift) | 0.73 | 0.94 | 0.67 | 0.73 | 0.59 |
| A140 (drift) | 0.74 | 0.67 | 0.95 | 0.73 | 0.86 |
| T140 (noise) | 0.99 | 0.77 | 0.76 | 0.98 | 0.65 |
| EffCS (noise) | 0.78 | 1.00 | 0.69 | 0.77 | 0.61 |
| A140 (noise) | 0.78 | 0.70 | 1.00 | 0.77 | 0.91 |
| T140 (drift + noise) | 0.70 | 0.55 | 0.55 | 0.69 | 0.47 |
| EffCS (drift + noise) | 0.72 | 0.93 | 0.65 | 0.71 | 0.58 |
| A140 (drift + noise) | 0.73 | 0.66 | 0.94 | 0.72 | 0.85 |

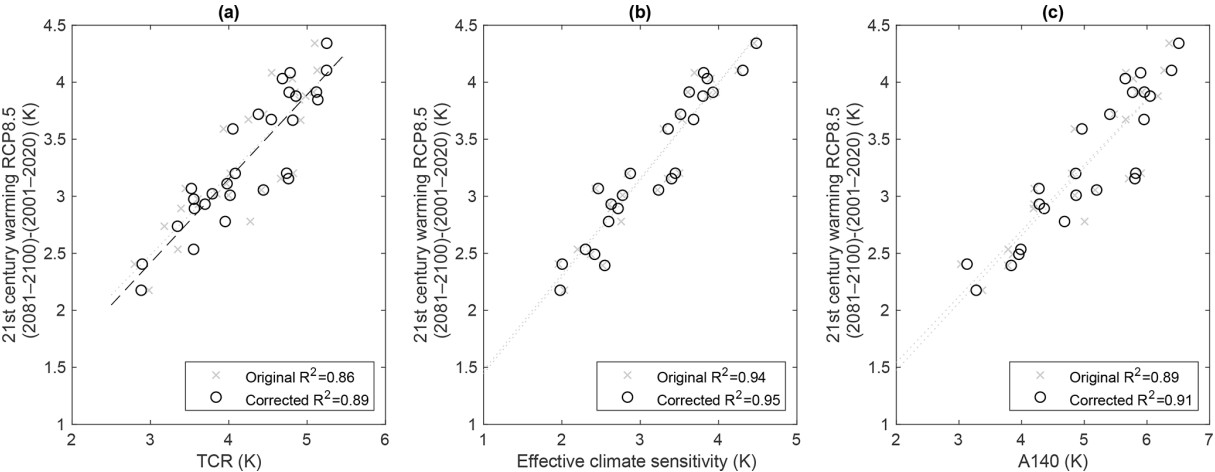

**Figure 6.** Plots showing the correlation between TCR **(a)**, EffCS **(b)** and A140 **(c)** with 21st century warming, here represented by the difference between 2001–2020 and 2081–2100 global mean temperatures in the first ensemble member for each model in the CMIP5 archive for the RCP8.5 scenario. Each plot shows the "original" calculation, where the baseline temperatures (and TOA fluxes for EffCS) are taken as the piControl mean. In the "corrected" calculation, a correction term for the baseline temperature and control drift is applied. Correlation coefficients are shown for the original and corrected cases.

## 4 Conclusions

The question of which metric of climate sensitivity is most useful for summarizing uncertainty in future projections is conditional on a number of factors. Any single metric of sensitivity, even if known perfectly, cannot constrain Earth system response on all timescales and scenarios. We have shown here that one can produce a number of model variants which can exhibit the same value of EffCS or TCR but with a range of responses, especially in a mitigation scenario such as RCP2.6.

In an idealized environment where models can be brought to a complete equilibrium control state, and ensemble sizes for "1pctCO2" simulations are large enough to avoid the effects of internal variability, the T140 metric would be the best idealized warming measure for century-scale warming under a high emissions scenario. However, the presence of even moderate control drift can act as a significant source of error in the measurement of T140, and so here we find that EffCS is likely to be a more accurate practical sensitivity metric in Earth system model applications where full equilibration is difficult to achieve.

https://doi.org/10.5194/esd-11-1-2020                    Earth Syst. Dynam., 11, 1–15, 2020

**Table 3.** A table showing various sensitivity metrics estimated from the CMIP5 and CMIP6 ensembles (in K), using both pre-industrial average baseline temperatures (org) and baseline temperatures estimated from a regression fit to the first 20 years of the 1cptCO2 simulation (corr), where the linear fit is used to estimate temperatures and radiative fluxes at $t = 0$. Warming is shown (where available) for corresponding RCP2.6 and RCP8.5 simulations, where the difference between 2001–2020 and 2081–2100 in the first ensemble member for the corresponding model is used to assess 21st century warming.

| Model | EffCS (org) | EffCS (corr) | A140 (org) | A140 (corr) | T140 (org) | T140 (corr) | RCP8.5 2000–2100 | RCP2.6 2000–2100 |
|---|---|---|---|---|---|---|---|---|
| ACCESS1_0 | 3.48 | 3.53 | 5.48 | 5.60 | 4.45 | 4.57 | 3.72 | – |
| ACCESS1_3 | 3.30 | 3.38 | 4.84 | 5.02 | 3.93 | 4.11 | 3.59 | – |
| BNU_ESM | 3.86 | 3.80 | 6.17 | 6.05 | 4.98 | 4.86 | 3.88 | 0.63 |
| CCSM4 | 2.84 | 2.87 | 4.80 | 4.86 | 4.02 | 4.08 | 3.20 | 0.44 |
| CESM1_CAM5_1_FV2 | 3.31 | 2.89 | 5.29 | 4.44 | – | – | – | – |
| CNRM_CM5 | 3.22 | 3.28 | 5.17 | 5.30 | 4.42 | 4.54 | 3.06 | 0.68 |
| CNRM_CM5_2 | 3.37 | 3.37 | 5.11 | 5.12 | 4.29 | 4.29 | – | – |
| CSIRO_Mk3_6_0 | 3.53 | 3.63 | 5.66 | 5.86 | 4.25 | 4.45 | 3.67 | 1.09 |
| CanESM2 | 3.61 | 3.59 | 5.92 | 5.89 | 5.08 | 5.05 | 3.91 | 0.92 |
| FGOALS_s2 | 3.85 | 3.78 | 5.90 | 5.76 | 4.76 | 4.62 | – | – |
| GFDL_CM3 | 3.69 | 3.87 | 5.66 | 6.02 | 4.55 | 4.90 | 4.08 | 1.25 |
| GFDL_ESM2G | 2.37 | 2.34 | 3.86 | 3.80 | – | – | 2.49 | −0.08 |
| GFDL_ESM2M | 2.52 | 2.60 | 3.78 | 3.93 | – | – | 2.39 | 0.32 |
| GISS_E2_H | 2.20 | 2.42 | 3.79 | 4.23 | 3.35 | 3.79 | 2.53 | 0.36 |
| GISS_E2_R | 2.03 | 2.01 | 3.37 | 3.34 | 2.98 | 2.94 | 2.18 | 0.09 |
| HadGEM2_ES | 4.25 | 4.34 | 6.27 | 6.45 | 5.13 | 5.30 | 4.10 | 0.87 |
| IPSL_CM5A_LR | 3.90 | 3.92 | 5.78 | 5.78 | 4.81 | 4.81 | 4.03 | 0.80 |
| IPSL_CM5A_MR | 3.96 | 4.01 | 5.84 | 5.93 | 4.84 | 4.93 | 3.91 | 0.59 |
| IPSL_CM5B_LR | 2.43 | 2.54 | 4.20 | 4.43 | 3.45 | 3.67 | 3.07 | – |
| MIROC_ESM | 4.45 | 4.51 | 6.35 | 6.56 | 5.10 | 5.30 | 4.34 | 1.26 |
| MIROC5 | 2.60 | 2.62 | 4.20 | 4.27 | 3.61 | 3.68 | 2.93 | 0.62 |
| MPI_ESM_LR | 3.50 | 3.45 | 5.91 | 5.82 | 4.82 | 4.74 | 3.20 | 0.43 |
| MPI_ESM_MR | 3.35 | 3.42 | 5.71 | 5.84 | 4.66 | 4.80 | 3.15 | 0.36 |
| MPI_ESM_P | 3.34 | 3.31 | 5.71 | 5.64 | 4.57 | 4.49 | – | – |
| NorESM1_M | 2.63 | 2.68 | 4.19 | 4.29 | 3.39 | 3.49 | 2.89 | 0.55 |
| bcc_csm1_1 | 2.77 | 2.77 | 4.85 | 4.87 | 4.00 | 4.02 | 3.01 | 0.52 |
| bcc_csm1_1_m | 2.76 | 2.68 | 5.00 | 4.84 | 4.27 | 4.11 | 2.78 | 0.30 |
| inmcm4 | 1.96 | 2.00 | 3.03 | 3.13 | 2.80 | 2.89 | 2.41 | – |
| BCC_CSM2_MR | 2.87 | 2.91 | 4.75 | 4.89 | 3.88 | 4.02 | – | – |
| CESM2 | 4.70 | 4.80 | 7.20 | 7.40 | 5.01 | 5.20 | – | – |
| CESM2_WACCM | 4.32 | 4.43 | 6.51 | 6.74 | 4.68 | 4.91 | – | – |
| GISS_E2_1_G | 2.61 | 2.66 | 4.18 | 4.27 | 1.95 | 2.04 | – | – |
| GISS_E2_1_H | 2.99 | 3.09 | 4.94 | 5.13 | 4.11 | 4.31 | – | – |
| MIROC6 | 2.40 | 2.40 | 3.96 | 3.98 | 3.47 | 3.49 | – | – |
| CNRM_CM6_1 | 4.69 | 4.67 | 6.75 | 6.71 | 5.49 | 5.46 | – | – |
| CNRM_ESM2_1 | 4.35 | 4.30 | 6.16 | 6.07 | 5.12 | 5.02 | – | – |
| UKESM1_0_LL | 5.26 | 5.14 | 7.66 | 7.41 | 6.36 | 6.11 | – | – |
| E3SM_1_0 | 5.26 | 4.68 | – | – | – | – | – | – |

EffCS itself has limitations; it is relatively insensitive to slow timescale feedbacks, which means that it poorly correlated with century-scale warming under RCP2.6 (where a large fraction of warming occurs due to slow feedback response to historical emissions) and for warming on multi-century timescales under a high emissions scenario (where concentrations stabilize post-2100). We find that a simple but useful alternative is to simply use the mean warming from years 131 to 150 of the abrupt4xCO2 simulation – which is

skilled comparably to EffCS in predicting RCP8.5 warming in 2100 but more sensitive to century timescale feedbacks than EffCS – therefore, it is better correlated with RCP2.6 end-of-century warming.

It is notable that the most common metrics of sensitivity (EffCS, T140 and TCR) provide very little guidance on peak warming expected under climate mitigation. The focus on these metrics has also given rise to the issue that slow feedbacks in Earth system models are not well constrained

by the set of experiments currently conducted by default in CMIP. The standard 150-year simulation used to calculate effective climate sensitivity does not constrain true equilibrium climate sensitivity, and only a limited set of CMIP-class models have run models for long enough to be informative about equilibrium response (Rugenstein et al., 2020).

It should be noted that these conclusions are derived from the consideration of a relatively simple two-timescale pulse response model. In this model, we can show that certain sensitivity metrics are insufficient to constrain future projections, and that non-equilibration may confound measurement. However, the constrained distributions for the metrics are subject to the structural assumptions of the model. The real world may have more than two response timescales (Aengenheyster et al., 2018) or may be better described as a continuous sum (Ragone et al., 2015; Lembo et al., 2020). Further work should identify how such complexity impacts uncertainty in relevant climate metrics.

The diversity of simulated global mean dynamical response to greenhouse gas forcing over the coming centuries can be represented in simple models with a relatively small number of parameters (Smith et al., 2018; Meinshausen et al., 2011), but we cannot reduce uncertainty in climate projections on all timescales to a single degree of freedom. Summary metrics of climate response have value if the context of those metrics (and their range of applicability in relation to projection uncertainty) is well understood, but their limitations should be kept in mind.

**Data availability.** CMIP5 and CMIP6 data are available through a distributed data archive developed and operated by the Earth System Grid Federation (ESGF).

**Code and data availability.** Code for this study is available on GitHub at https://doi.org/10.5281/zenodo.3835542 (Sanderson, 2020).

**Supplement.** The supplement related to this article is available online at: https://doi.org/10.5194/esd-11-1-2020-supplement.

**Competing interests.** The author declares that there is no conflict of interest.

**Acknowledgements.** This work is funded by the French National Research Agency, project no. ANR-17-MPGA-0016. Benjamin Sanderson is an affiliate scientist with the National Center for Atmospheric Research, sponsored by the National Science Foundation.

**Financial support.** This research has been supported by the Agence Nationale de la Recherche (grant no. ANR-17-MPGA-0016).

**Review statement.** This paper was edited by Valerio Lucarini and reviewed by two anonymous referees.

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

## Remarks from the typesetter