# Peer review of "Relating Climate Sensitivity Indices to projection uncertainty"

_Earth System Dynamics, 2019_

## Referee Comment (RC1) · Anonymous Referee #1 · 13 Jan 2020

In the present paper the author assesses the relation of different traditional metrics of climate sensitivity to future warming in a simple climate model framework. Ensemble simulations for RCP8.5 and RCP2.6 scenarios with a two timescale thermal response model constrained by observations are conducted. Two commonly used (Effective Climate Sensitivity (EFFCS) and Transient Climate Response (TCR/T140)), and one new proposed (A140) metrics are assessed. The findings are discussed in relation to CMIP5 and CMIP6 data. The study indicates that sensitivities derived from different metrics are time-scale dependent. Residual drift in the control run can substantially affect the significance. In particular, drift may explain that, surprisingly, EffCS is a better predictor than TCR for CMIP RCP8.5 simulations.

General

[Figure]

Simple metrics like EFFCS and TCR are frequently used to, e.g., assess the climate sensitivity of Earth system models. On the other hand, various studies have documented limitations of such simple metrics, which may lead to erroneous conclusions. Thus, assessing the applicability of such metrics is a valuable contribution. A simple model framework, as used in this study, may by an appropriate testbed providing that the applicability of the results can be conclusively demonstrated. In my view, the present study presents some interesting and valuable results. Overall, it is well-written (though the model description may need some improvement, see below), and well-structured. However, I think that major modifications are needed to justify publication. In particular, with regard to a more comprehensive and clear description of the model (see Major 1 & 2), and with regard to the applicability of the results (see Major 3 & 4). In addition, the author may consider few minor points to further improve the paper.

Major

1) I have problems to completely understand the simple model setup. A more thorough description is needed in my view. As far as I understand, the model consists of Eqs. B1 and B2, together with B3 for including a transient forcing. An optimization procedure is applied to estimate the parameters based on a given data set (HadCRUT) and cost functions (B4-B6). However, I do not completely understand how this optimization defines the parameter distribution (the model ensemble), i.e. how is the distribution exactly derived from the optimization, and how do(es) the distribution(s) look(s) like (a Figures of the pdfs may be helpful in this respect)? Furthermore (random order): (i) in L210 it is stated that $CO_2$ concentrations enter the cost functions, but $H(t)$ and $D(t)$ seem to be heat fluxes (L215)? (ii) How do $H$ and $D$ relate to the parameters $r_1, r_2$ in B2 (are they the same)? (iii) Are $T_p$ (B3) and $P$ (B1) the same? (iv) How does $F$ (B3) relate to $R$ (B2) (or how are eq. B1 and eq. B2 coupled in the model). (v) How (where) does the non $CO_2$ forcing factor $f_r$ (L208) enters the equations.

2) I'm wondering how important the non-$CO_2$ forcing agents (L207) and the factor $f_r$ are for the results. How much of the variability of the control (present day) climate is

explained by the non-CO2 forcing, and how is the non-CO2 forcing prescribed in the scenarios (it seems that all is represented by a constant $f_r$)?

3) Not much attempt is made to evaluate/validate the models behaviour under RCP scenarios. So far (as far as I can see) it is only shown that the model reasonably reproduces the HadCRUT data (where it is constraint to), and gives a response within the CMIP range. It would be useful to show that the model can reasonably reproduce the RCP8.5/RCP2.6 response of one particular model if the parameters are constraint by the present day simulation of the same model. This would give more confidence to the obtained results.

4) One main result is that residual drift may explain 'surprising' results regarding EffCS and TCR in CMIP. From Fig. 3 we see that different CMIP models seem to exhibit different magnitudes of residual drift. I'm wondering whether the simple model result regarding the effect of drift can be qualitatively checked by comparing respective simulations.

Some Minor

1) A common question concerning studies utilizing such simplified models is the sensitivity to the particular choice of the model setup. In this respect, the author may like to comment on the sensitivity of the results with respect to the particular choice of the number of timescales (n=2 in B1 & B2). How different would be the results for n=1 (or n=3)?

2) The author introduces a new metric (A140) as an alternative. It would be useful if the author could illustrate the behaviour of A140 (in contrast to EFFCS) for a CMIP data set.

3) In the abstract, the author quantifies the relative errors for T140 and EFFCS in the simple model framework. As these numbers may certainly not be the same for CMIP model, the may not be part of the abstract.

4) f_r appears twice in Table B1

---

## Referee Comment (RC2) · Anonymous Referee #2 · 23 Jan 2020

The manuscript illustrates the main differences between metrics that are commonly used to evaluate the response to GHG forcing in climate models. Namely, these are the effective climate sensitivity (EffCS), the transient climate response at CO2 quadrupling (T140), transient climate response at CO2 doubling (TCR), the temperature change after 140 years from CO2 quadrupling (A140). A simple impulse-response model is introducing, separating the response into a fast and a slow component, and whose model parameters are a posterior evaluated through minimization of the cost function with respect to an observational-based dataset. This model is used to consider to what extent the mentioned metrics are able to explain the response to a business-as-usual (RCP8.5) and mitigation (RCP2.6) scenario. It is found that different metrics are able to explain the response to different forcings, and that the simple model that is

here proposed provides different results, compared to state-of-the-art climate models from CMIP5 Project. It is argued that the biases affecting model energy conservation ultimately affect the different explaining capability of the CMIP5 models.

General comment:

Overall, I think that the manuscript is well written, the issue has a great scientific relevance, and the arguments here shown provide significant advancement to the discussion on the topic. Thus, I appreciate that the author addresses them critically, emphasizing that their adoption is conditioned to the problem that one needs to focus on. This is in line with previous works having evidenced the limitations of these metrics for the study of the climate response, especially from a modelling perspective.

I am a bit skeptical about the effectiveness of the impulse-response model, given that it is a purely linear context. The addition of the noise+drift, though, is convincing in explaining part of the discrepancy between the simple model and CMIP5 outputs. The arguments about the applicability of the metrics are thus promising also in a "real-world" context (using the notation adopted by the author), although with some caveats. For this reason, I think it is important that the author puts more emphasis on the nature of the impulse-response model, in the framework of linear response theory (LRT) and Hasselmann-type response (see my specific comments), and evidences its limits.

I think some improvements can be made in terms of how the methodology and results are described. It would be useful to have the "Methods" section in the main part of the manuscript, instead of as an appendix. The notation is not always consistent across the methods. The MCMC procedure should be explicitly described, not only by mentioning the original reference. Some figure captions lack important details.

Specific comments in the following are meant to clarify my general comments and constrain them to the relative sections of the manuscript. I have spotted several typos; these are addressed in the minor comments.

Specific comments:

- ll. 90-92: I do not have clear how the normalized regression coefficients shown in Figure 1f support this argument. Can the author better clarify it?

- ll. 94-96: I do not understand this sentence for a few reasons. Firstly, is the author referring to any specific forcing, when he says that the rate of change in the forcing is approximately constant? In the case of the mitigation scenario (RCP2.6), this is obviously not the case. Secondly, I do not have clear in mind what the author means by "saturation" of the fast feedback response, and if this refers to the whole period 2000 to 2100 or to the end of the period.

- l. 132: the author suggests that the contributions of the two factors are separately addressed in the following, but, in the end, only the overall effect of the bias is taken into account in the following.

- ll. 161-162: the author seems to imply that "real-world applications" are prone to the existence of drifts. But this is rather a model issue, as the unforced "real-world" climate system should not have any drift.

- l. 166: the author did not specify anywhere else in the text what is the length of the abrupt 4xCO2 simulation. As a consequence, "end" of the simulation does not seem to have a specific meaning.

- Appendix B: according to ESD standards, I think that it would be more appropriate if the Methods section are moved in the main text after the Introduction. Moreover, a description of the data that have been used is lacking, especially for what concerns the observational-based datasets used for model optimization.

- Eqs. B1-B2: the impulse response model here adopted requires using only two timescales. Is it sufficient to describe the response? The FAIR impulse-response model here mentioned includes a set of four simple feedback equations (cfr. Hasselmann et al. 1993) differing on the magnitude of the feedback parameter (i.e. on the
timescale of the response). What happens if one includes more than the two timescales considered in this analysis, given that similar strategies applied to geoengineering scenarios have used, for instance, three exponentials (cfr. Aengenheyster et al. 2018)? This is particularly relevant, as the impulse-response model can always be expressed as an infinite sum of exponential behaviors, differing in their timescale, but the response of the real system rarely has the shape of a discrete number of exponential behaviors combined with each other (e.g. Ragone et al. 2016: Lembo et al. 2019). Also, the adoption of the fast-slow scale implies a separation of scale, that is here inferred "a posterior" through heuristic arguments. Nevertheless, there is no reason, in principle, to assume that a scale separation exists, and this problem traces back to the very foundations of the theory about climate response and forced-free fluctuations dichotomy (Lorenz 1979). One way to deal with that would be to evaluate the memory term (cfr. Ghil and Lucarini 2019). I understand that this might go beyond the scope of this work, but I wonder if the author might comment on that in the manuscript.

- Eq. B3: according to the convolution properties, this operation is by all means equivalent to the application of the Ruelle Response Theory (RRT) (Ruelle 1998a; Ruelle 1998b) when a hypothetical impulse perturbation is applied, allowing for a particularly simple derivation of the linear Green function (cfr. Hasselmann et al. 1993). This has found several applications in the context of climate prediction (cfr. Ragone et al. 2016; Lucarini et al. 2017; Ghil and Lucarini, 2019 for a review), not only constraining to the temperature response, but also to a wide range of climatic variables (e.g. Helwegen et al. 2019; Lembo et al. 2019). These arguments provide a rigorous mathematical framework to the experimental protocol here described.

- Sect. B1.1: I believe that a complete description of the model is here lacking and should be included. Referring to the model settings, in particular, it is not clear to me how the ensemble is generated and how many members are taken into account.

- Table B1: is it possible to have a range for rn as well? Also, where does the fr parameter enter the mode? This goes back to my minor comment about consistent

notation.

- l. 226: I think that it is important to notice here that in the forcing scenario 1pctCO2 the CO2 concentration reaches doubling after 70 years, as I presume that this motivates the choice of the 61-80 and 131-150 20-years averages.

- Figure A1: the caption does not contain an explanation of the panel b content. Particularly, the author might want to explain the meaning of the red shading, and the range encompassed by the dotted lines.

- Figure A2: the author does not explain why the choice of a single member from each CMIP model ensemble is reasonable in this context.

- Figure A3: it appears that the distributions of fast-scale parameters are much more similar to a Gaussian distribution, compared to the slow-scale parameters. I am surprised that the author does not refer to that explicitly and comments on it. Could it be an evidence that the scale separation that is a priori assumed for parameter model optimization is such that the fast-scale system approaches a stochastic process, in the context of the response of the system to the impulse forcing? This would be certainly reasonable, in an "Hawkins and Sutton, 2009 context" (signal-to-noise ratio approach), but the author might want to justify it in a more rigorous way.

Minor comments:

- l. 19: in this sentence there is a repetition ("range" and "ranging"). Consider rearranging the sentence;

- l. 31-32; I found this part of the sentence a bit difficult to read. A suggestion might be to replace it with "a complication has arisen due to the fact that EffCS seems to be better correlated than TCR with 21st Century warming from present day levels under a business-as-usual scenario."

- l. 37: replace "have" with "of".

- l. 60: remove "to".

- l. 66: it is not clear whether the author refers here to the Appendix A, Appendix B or both.

- l. 69: replace "and" with "to".

- l. 91: either a sentence breaking is needed here (after the brackets), or "suggest" has to be replaced by "suggesting".

- l. 125: Replace "of CMIP5" with "for CMIP5".

- l. 138: if the "Methods" section is in the appendix, they have to be referred to more appropriately as "Appendix (B)".

- l. 147: replace "the both" with "both".

- l. 151: replace "Supplemental" with "Supplementary".

- l. 165: replace "an" with "that a".

-Eq. B3: I noticed a potential mismatch in the notation, compared to eq. B1. The author may consider adopting the same notation for the temperature evolution in both equations.

- Figure A1: replace "senstivity" with "sensitivity" in the caption.

---

## Author Response (AR1)

**Response to Reviewer 1**

Thanks to the reviewer for thorough reading and thoughtful points. I've endeavoured to address the issues raised in the revised paper as detailed below.

**Major Issues**

*1)... the model consists of Eqs. B1 and B2, together with B3 for including a transient forcing. An optimization procedure is applied to estimate the parameters based on a given data set (HadCRUT) and cost functions (B4-B6). I do not completely understand how this optimization defines the parameter distribution (the model ensemble),*

I have expanded the discussion of what the MCMC optimization does and how the parameter distribution is derived from the cost function

*i.e. how is the distribution exactly derived from the optimization, and how do(es) the distribution(s) look(s) like (a Figures of the pdfs may be helpful in this respect)?*

The relevant figure is in the supplemental material, Figure S3 - which shows the individual and pairwise parameter distributions of the posterior ensemble.

*Furthermore (random order):*

*(i) in L210 it is stated that CO2 concentrations enter the cost functions, but H(t) and D(t) seem to be heat fluxes (L215)?*

Sorry - that was a typo, the treatment of CO2 is specific to the companion paper to this one, which included carbon cycle feedbacks. This paper only considers the thermal part of the model. Typo corrected.

*(ii) How do H and D relate to the parameters r1,r2 in B2 (are they the same)?*

Now included specific equations for D and H

*(iii) Are T_p (B3) and P (B1) the same?*

Yes, now T_p throughout

*(iv) How does F (B3) relate to R (B2) (or how are eq. B1 and eq. B2 coupled in the model).*

The coupling is in the original multi-timescale energy balance model as detailed in Millar et al 2017 (now written explicitly). The particular solutions of the temperature and radiative response to a step change in forcing can be written as a sum of exponential decays (again, now shown explicitly)

*(v) How (where) does the non CO2 forcing factor f_r (L208) enters the equations.*

Now included an explicit expansion of the historical forcing timeseries F(t) which defines f_r

*2) I'm wondering how important the non-CO2 forcing agents (L207) and the factor f_r are for the results. How much of the variability of the control (present day) climate is explained by the non-CO2 forcing, and how is the non-CO2 forcing prescribed in the scenarios (it seems that all is represented by a constant f_r)?*

The constant f_r is a scaling factor (as now made clear by equation B5), so the non-CO2 forcing is not constant over time - but you are correct that we are assuming that there is only a single degree of freedom in optimization. Though we could break down this forcing further - our primary goal is not to attribute the response to different forcings, and this formulation allows conceptually for uncertainty in the historical forcing timeseries while minimizing the number of degrees of freedom in the optimization.

*3) Not much attempt is made to evaluate/validate the models behaviour under RCP scenarios. So far (as far as I can see) it is only shown that the model reasonably reproduces the HadCRUT data (where it is constrained to), and gives a response within the CMIP range. It would be useful to show that the model can reasonably reproduce the RCP8.5/RCP2.6 response of one particular model if the parameters are constrained by the present day simulation of the same model. This would give more confidence to the obtained results.*

I've included an additional supplementary plot S4 to fit the pulse-response model to historical simulations in the CMIP archive with future ensemble projections for RCP2.6 and 8.5, with some discussion in the methods.  The technique generally performs well (i.e. future projections fall within the distribution), with the exception of a couple of models which show little or no long term warming response (CCSM4, FGoals, FIO-ESM - models which share some fraction of their codebase).  I've added a caveat to this effect, but I'm broadly happy that the technique is producing reasonable probabilistic fits to historical CMIP data.

*4) One main result is that residual drift may explain 'surprising' results regarding EffCS and TCR in CMIP. From Fig. 3 we see that different CMIP models seem to exhibit different magnitudes of residual drift. I'm wondering whether the simple model result regarding the effect of drift can be qualitatively checked by comparing respective simulations.*

I've attempted to show this qualitatively in a new Figure 5, and supplementary Figure S5.  The former is an attempt to 'correct' the control drift uncertainty in the estimation of TCR by estimating baseline temperatures from the 1pctCO2 simulation itself, and background trends from the control simulation.  The plots show - for all 3 metrics, but particularly for TCR - that the correlation with 21st century warming under RCP8.5 can be improved a little using this baseline correction, supporting the hypothesis that control drift is an issue for the estimation of sensitivity metrics.  Of course this is just an estimate - itself noisy given the relatively short regression,

which is noted in the text at the end of the results section, but the improvement over the PIControl average is notable.

*Some Minor*

*1) A common question concerning studies utilizing such simplified models is the sen- sitivity to the particular choice of the model setup. In this respect, the author may like to comment on the sensitivity of the results with respect to the particular choice of the number of timescales (n=2 in B1 & B2). How different would be the results for n=1 (or n=3)?*

Repeating the entire analysis with a different model dimensionality is beyond scope, but during development, I experimented with different timescales dimensions - 1 timescale can be trivially dismissed as unable to represent the temporal evolution of the models in response to 4xCO2 forcing.  Beyond two timescales, only slight improvement is seen in the fitting error - so two timescales was chosen for this study to be (a) consistent with existing literature (i.e. within the framework of FAIR, which is in common usage), (b) lower dimensional so easier to interpret in terms of slow/deep ocean and fast/shallow ocean response and (c) applicable to CMIP models in.

Fundamentally - only some models show a slightly improved fit with an extra allowed timescale (see GISS-H, for example on the below plot).  Other models are adequately described with 2, and adding a 3rd results in a degenerate fit.  Thus - a cross ensemble analysis of the additioanl degree of freedom is not clearly defined.  Other studies have arrived at the same conclusion (see Proistosescu and Huybers http://doi.org/10.1126/sciadv.1602821,  Smith 2018 http://dx.doi.org/10.5194/gmd-11-2273-2018 , Geoffroy 2012 http://doi.org/10.1175/JCLI-D-12-00196.1 ).

Ultimately, for this study, the aim is to reproduce the basic features of CMIP ensemble diversity in response to different types of forcing with the minimum possible complexity of model - and I felt that this was both possible and easier to explain with the two timescale model.  Clearly, the real world could  have the capacity to respond to forcing on a range of timescales, but two timescales adequately describe the response to forcing on the century timescale in the CMIP ensemble.

[Figure]

I have added a paragraph in the conclusions on how the structural assumption of two timescales may impact results. Primarily - I think the current method is support the primary conclusions that ECS and TCR are insufficient to constrain some future warming trends (i.e. ECS or TCR are insufficient to describe RCP2.6 warming), and that non-equilibration is a problem for measurement.  But I do agree that the structural assumption of 2 timescales might impact the constraint, for example, of ECS from historical temperatures - and I've added this caveat.

*2) The author introduces a new metric (A140) as an alternative. It would be useful if the author could illustrate the behaviour of A140 (in contrast to EFFCS) for a CMIP data set.*

A140 is included in new Figure 5, and estimated values are included in new Table 2.

*3) In the abstract, the author quantifies the relative errors for T140 and EFFCS in the simple model framework. As these numbers may certainly not be the same for CMIP model, the may not be part of the abstract.*

I have removed these quantitative results from the abstract accordingly

*4) f_r appears twice in Table B1*

**Response to Reviewer 2**

*Overall, I think that the manuscript is well written, the issue has a great scientific rele- vance, and the arguments here shown provide significant advancement to the discus- sion on the topic. Thus, I appreciate that the author addresses them critically, empha- sizing that their adoption is conditioned to the problem that one needs to focus on. This is in line with previous works having evidenced the limitations of these metrics for the study of the climate response, especially from a modelling perspective.*

Many thanks for the positive evaluation and careful reading.

*I am a bit skeptical about the effectiveness of the impulse-response model, given that it is a purely linear context. The addition of the noise+drift, though, is convincing in explaining part of the discrepancy between the simple model and CMIP5 outputs. The arguments about the applicability of the metrics are thus promising also in a "real- world" context (using the notation adopted by the author), although with some caveats. For this reason, I think it is important that the author puts more emphasis on the nature of the impulse-response model, in the framework of linear response theory (LRT) and Hasselmann-type response (see my specific comments), and evidences its limits.*

These points are well taken - and thanks to the reviewer for the additional literary context, which I've endeavoured to include.  I've tried to put the two timescale model in appropriate context - the primary defense for this application being that it is already sufficiently complex to show that TCR and ECS do not constrain future warming under strong mitigation, and that non-equilibration is a potential issue for TCR estimation.  I believe that these points, which are statements of lack of confidence, are robust to the consideration of a wider set of models with additional response timescales.

I do however agree that the 2-timescale structural assumption is strong - and any constrained distribution (of future warming, EffCS or TCR) need to be considered in the context of these caveats.  For this reason, I do not highlight the actual constrained ranges here - and I have added an additional paragraph to the conclusions to explain this.

*I think some improvements can be made in terms of how the methodology and results are described. It would be useful to have the "Methods" section in the main part of the manuscript, instead of as an appendix.*

I have restructured the document to have the methods in line.

*The notation is not always consistent across the methods.*

I've worked to reformat the methods extensively following the comments by both reviewers

*The MCMC procedure should be explicitly described, not only by mentioning the original reference.*

I've included an extended description of the algorithm and the reasons for using it.

**Specific comments:**

*- ll. 90-92: I do not have clear how the normalized regression coefficients shown in Figure 1f support this argument. Can the author better clarify it?*

I've deleted this paragraph - as I think the point is overly subtle.

*- ll. 94-96: I do not understand this sentence for a few reasons. Firstly, is the author referring to any specific forcing, when he says that the rate of change in the forcing is approximately constant? In the case of the mitigation scenario (RCP2.6), this is obviously not the case.*

Apologies - this paragraph was talking explicitly about RCP8.5, in which total radiative forcing increases broadly linearly throughout the 21st century. I've rewritten this section.

*Secondly, I do not have clear in mind what the author means by "saturation" of the fast feedback response, and if this refers to the whole period 2000 to 2100 or to the end of the period.*

Section now deleted.

*- l. 132: the author suggests that the contributions of the two factors are separately addressed in the following, but, in the end, only the overall effect of the bias is taken into account in the following. -*

Thanks - corrected. I now come back to the unknown baseline factor in the CMIP detrending exercise at the end of the results section.

*ll. 161-162: the author seems to imply that "real-world applications" are prone to the existence of drifts. But this is rather a model issue, as the unforced "real-world" climate system should not have any drift.*

Corrected.

*- l. 166: the author did not specify anywhere else in the text what is the length of the abrupt 4xCO2 simulation. As a consequence, "end" of the simulation does not seem to have a specific meaning.*

Replaced by "years 121-140"

*- Appendix B: according to ESD standards, I think that it would be more appropriate if the Methods section are moved in the main text after the Introduction.*

Done - methods are now inline in the text

*Moreover, a description of the data that have been used is lacking, especially for what concerns the observational-based datasets used for model optimization.*

All relevant citations are now included.

*- Eqs. B1-B2: the impulse response model here adopted requires using only two timescales. Is it sufficient to describe the response? The FAIR impulse-response model here mentioned includes a set of four simple feedback equations (cfr. Hassel- mann et al. 1993) differing on the magnitude of the feedback parameter (i.e. on the timescale of the response). What happens if one includes more than the two timescales considered in this analysis, given that similar strategies applied to geoengineering sce- narios have used, for instance, three exponentials (cfr. Aengenheyster et al. 2018)?*

I fully agree that 2 timescales is a structural assumption, and that additional timescales of response would be likely required for longer periods of response. During development, I experimented with different timescales dimensions - 1 timescale can be trivially dismissed as unable to represent the temporal evolution of the models in response to $4xCO_2$ forcing. Beyond two timescales, only slight improvement is seen in the fitting error - so two timescales was chosen for this study to be (a) consistent with existing literature (i.e. within the framework of FAIR, which is in common usage), (b) lower dimensional so easier to interpret in terms of slow/deep ocean and fast/shallow ocean response and (c) sufficient for demonstrating the main point that drift and noise impact TCR more than ECS.

For 140 abrupt-$4xCO_2$ response, only some models show an improved fit with an extra allowed timescale (see GISS-H, for example on the below plot), and even then it's a slight improvement. Most models are adequately described with 2, and adding a 3rd results in a degenerate fit. Other studies have arrived at the same conclusion for summarizing responses on the century timescale (see Proistosescu and Huybers http://doi.org/10.1126/sciadv.1602821, Smith 2018 http://dx.doi.org/10.5194/gmd-11-2273-2018 , Geoffroy 2012 http://doi.org/10.1175/JCLI-D-12-00196.1 ).

Ultimately, for this study, the aim is to reproduce the basic features of CMIP ensemble diversity in response to different types of forcing with the minimum possible complexity of model - and I felt that this was both possible and easier to explain with the two timescale model. Clearly, the real world could have the capacity to respond to forcing on a range of timescales, but two timescales adequately describe the response to forcing on the century timescale in the CMIP ensemble.

[Figure]

*This is particularly relevant, as the impulse-response model can always be expressed as an infinite sum of exponential behaviors, differing in their timescale, but the re- sponse of the real system rarely has the shape of a discrete number of exponential behaviors combined with each other (e.g. Ragone et al. 2016: Lembo et al. 2019). Also, the adoption of the fast-slow scale implies a separation of scale, that is here in- ferred "a posterior" through heuristic arguments. Nevertheless, there is no reason, in principle, to assume that a scale separation exists, and this problem traces back to the very foundations of the theory about climate response and forced-free fluctuations dichotomy (Lorenz 1979). One way to deal with that would be to evaluate the memory term (cfr. Ghil and Lucarini 2019). I understand that this might go beyond the scope of this work, but I wonder if the author might comment on that in the manuscript.*

This point is well taken - though to redesign the model as an infinite sum would create a challenge in terms of a low-dimensional parametric definition which could be used in MCMC. However - I recognise that the discrete response assumption is a strong one, and I've added a paragraph in the discussion to outline this caveat in the interpretation of the results.

> *" These conclusions are derived from the consideration of a relatively simple two-timescale pulse response model which is sufficient to show that constraining certain types of sensitivity metric is insufficient to*

*constrain future projections, and that non-equilibration may confound
measurement, however, the constrained distributions for the metrics are
subject to the structural assumptions of the model used. The real world
may have more than two response timescales (Aengenheyster 2018), or
may be better described as a continuous sum (Ragone 2016, Lembo
2019). Further work should identify how such complexity impacts
uncertainty in relevant climate metrics."*

*- Eq. B3: according to the convolution properties, this operation is by all means equiv- alent to
the application of the Ruelle Response Theory (RRT) (Ruelle 1998a; Ruelle 1998b) when a
hypothetical impulse perturbation is applied, allowing for a particularly simple derivation of the
linear Green function (cfr. Hasselmann et al. 1993). This has found several applications in the
context of climate prediction (cfr. Ragone et al. 2016; Lucarini et al. 2017; Ghil and Lucarini,
2019 for a review), not only constraining to the temperature response, but also to a wide range
of climatic variables (e.g. Helwegen et al. 2019; Lembo et al. 2019). These arguments provide a
rigorous mathematical framework to the experimental protocol here described.*

Thanks for these. I've included the references when introducing the model.

*- Sect. B1.1: I believe that a complete description of the model is here lacking and should be
included. Referring to the model settings, in particular, it is not clear to me how the ensemble is
generated and how many members are taken into account.*

This section has been significantly expanded, and now includes a perfect model demonstration
fitting the model to CMIP members.

*- Table B1: is it possible to have a range for rn as well? Also, where does the fr parameter enter
the mode? This goes back to my minor comment about consistent notation.*

This is now clarified in the text. $r\_1$ is varied ($r\_2$ is ($1-r\_1$) due to the initial boundary condition).
$F\_r$ is now explicitly detailed in Eqn. 5.

*- l. 226: I think that it is important to notice here that in the forcing scenario 1pctCO2 the CO2
concentration reaches doubling after 70 years, as I presume that this motivates the choice of
the 61-80 and 131-150 20-years averages.*

Now noted explicitly, thanks.

*- Figure A1: the caption does not contain an explanation of the panel b content. Partic- ularly,
the author might want to explain the meaning of the red shading, and the range encompassed
by the dotted lines.*

Expanded.

*- Figure A2: the author does not explain why the choice of a single member from each CMIP model ensemble is reasonable in this context.*

I've now noted that the plot is subject to internal variability, but this is a central point which is being made. I am not trying to assess what is the most robust sensitivity metric given a situation where there is noise and potentially drift in the simulations. To have a subset of models with large ensemble averages (and others without) would confuse that assessment.

*- Figure A3: it appears that the distributions of fast-scale parameters are much more similar to a Gaussian distribution, compared to the slow-scale parameters. I am surprised that the author does not refer to that explicitly and comments on it. Could it be an evidence that the scale separation that is a priori assumed for parameter model optimization is such that the fast-scale system approaches a stochastic process, in the context of the response of the system to the impulse forcing? This would be certainly reasonable, in an "Hawkins and Sutton, 2009 context" (signal-to-noise ratio approach), but the author might want to justify it in a more rigorous way.*

I've expanded this discussion a little - though I'm not sure that we can infer any dynamical separation of timescales from the differences in distribution. My interpretation is that the fast timescales are simply more strongly constrained by the observations, whereas there are solutions with a wide range of slow timescale responses.

*Minor comments:*

*- l. 19: in this sentence there is a repetition ("range" and "ranging"). Consider rearrang- ing the sentence;*

Thanks, corrected.

*- l. 31-32; I found this part of the sentence a bit difficult to read. A suggestion might be to replace it with "a complication has arisen due to the fact that EffCS seems to be better correlated than TCR with 21st Century warming from present day levels under a business-as-usual scenario."*

Thanks - corrected as suggested.

*- l. 37: replace "have" with "of".*

Thanks, corrected.
*- l. 60: remove "to"*

Sentence removed
*- l. 66: it is not clear whether the author refers here to the Appendix A, Appendix B or both.*

Methods are now inline with the paper.

*- l. 69: replace "and" with "to".*

Thanks, done

*- l. 91: either a sentence breaking is needed here (after the brackets), or "suggest" has to be replaced by "suggesting".*

Sentence removed.

*- l. 125: Replace "of CMIP5" with "for CMIP5".*

done

*- l. 138: if the "Methods" section is in the appendix, they have to be referred to more appropriately as "Appendix (B)".*

Methods now inline.

*- l. 147: replace "the both" with "both".*

Thanks, corrected

*- l. 151: replace "Supplemental" with "Supplementary".*

done

*- l. 165: replace "an" with "that a".*

done

*-Eq. B3: I noticed a potential mismatch in the notation, compared to eq. B1. The author may consider adopting the same notation for the temperature evolution in both equations.*

This is now consistent throughout.

*- Figure A1: replace "senstivity" with "sensitivity" in the caption.*

**List of Changes in Manuscript**

- Methodology moved to main section and expanded to provide more complete description of the model used
- Expanded discussion of Markov Chain optimization
- Added validation of technique using historical CMIP simulations and RCP2.6/8.5 projections (results shown in Figure S4)
- Tested effect of residual drift on TCR estimation (latter part of results section and Figure 5, and Figure S5)
- Added discussion of number of exponential modes in model
- Added new Table 2 illustrating sensitivity metrics for CMIP models
- Expanded literature review of pulse-response formulation & linear response theory

[revised manuscript text omitted]

**Appendix : Supplementary Material**

**Figure S2.** Scatterplots of 21st century warming (difference between 20 year means in 2081-2100 and 1981-2000) and a range of sensitivity metrics for CMIP5. TCR, T140 and EffCS are reported values from (Stocker et al., 2013), A140 is calculated as the year 131-150 average global mean temperature above the control level (taken as the last 100 years of the relevant control simulation). Columns represent different RCPs, rows represent different sensitivity metrics considered in the text. Each point represents a single model from the archive. Only results from the 1st initial condition ensemble member are considered for each model (thus the plots are subject to initial condition variability).

[Figure]

**Figure S3.** A 'corner-plot' showing the posterior parameter distribution attained by MCMC calibration of the simple climate model. Diagonal plots show posterior histograms for parameter values optimized in the calibration, while the horizontal range indicates the bounding values of the initial flat prior distribution. Off-diagonal plots show pairwise distributions of parameters in the posterior distribution.

[Figure]

**Figure S4.** A demonstration of the simple model fitting strategy applied to historical simulations for a range of models in the CMIP5 archive. A pulse-response model is fitted treating each model's global mean temperature output in turn as truth for the period 1870-2019 (black line). 10th-90th percentiles of fitted temperature response for historical (grey area) and future projections are shown for RCP8.5 (pink area) and RCP2.6 (blue area) concentration pathways. Dotted lines show the median temperature in the ensemble projection, while solid colored lines show the evolution of the actual GCM for the corresponding scenario.

[Figure]

**Figure S5.** Figure illustrating the 'correction' employed for TCR and ECS in Figure 5. Corrected baseline temperatures are estimated by regression of the first 20 years of the control simulation, and branch-point from the control simulation is identifying by finding the year in which a linear fit to the control model evolution intersects the corrected baseline temperature. Branching in cases where there is no intersection are illustrated by the year in which the trendline is closest to the corrected baseline (either the first or last year).

[Figure]

---

## Author Response (AR2)

**Relating Climate Sensitivity Indices to projection uncertainty**

Many thanks to the reviewers for their constructive reviews, which I address point-by-point in the following:
* * *
**Reviewer 1**

**Reviewer Point P 1.1** — The manuscript in the current version has substantially improved compared to the initial draft. The order of the sections is more suitable for the ESD format and a number of details have been added to clarify the methodology. Particularly, I appreciated that the authors now explicitly discuss the choice of the two timescales for the optimisation in section 1 and, again, in the conclusions. Still, not only Proistosescu and Huybers, 2017, but also other authors (Caldeira et al. 2013; Joos et al. 2013; Tsutsui 2017) have chosen the three timescales for a pulse-response exponential relaxation model, similarly through an educated guess based on heuristic considerations. Therefore, I wonder if the author might explicitly mention the motivation behind his choice, the same way he did in the response to the reviewers. Provided that the author accounts for this general comment, and for the specific suggestions listed in the following, I thereby recommend that the manuscript is accepted for publication.

**Reply**: Thanks, once again for a careful and very constructive review. Regarding this suggestion, I have included an additional paragraph in the methods sections when introducing the 2 timescale model, defending the logic in the same manner as used in the paper response. I've also included an additional supplemental figure 5 to illustrate the logic.

**Reviewer Point P 1.2** — - l. 66 and elsewhere: there are probably some LaTeX problems with the citation format for several references throughout the manuscript

**Reply**: Noted and fixed throughout..

**Reviewer Point P 1.3** — - Eq.2: the equations appear a bit packed here. Consider introducing a broader spacing

**Reply**: Done

**Reviewer Point P 1.4** — - ll. 87-88: as far as my memory goes, the value of 3.7 Wm-2 for the CO2 radiative forcing is computed at doubling compared to pre-industrial baseline (see IPCC AR4, sect. 2.3). Can the authors explain why they refer to it as for CO2 quadrupling?

**Reply**: Apologies - it should indeed read $7.4 Wm^{-2}$, thanks for catching that.

**Reviewer Point P 1.5** — - l. 174: this is the only explicit reference to Figure S3 in the manuscript. Nevertheless, this figure contains quite a relevant part of the information that this work aims at conveying. The author might consider moving it from the supplementary material, or at least attach it to a more extensive discussion, since it is crucial in order to support the discussion on the role of the historical forcing for the role of the metrics in the CMIP models (ll. 192-198);

**Reply**: I've put the revised corner plot in the main text as suggested and expanded the discussion of the constrained MCMC posterior distribution

**Reviewer Point P 1.6** — - Figure S4: I think this figure might benefit by reducing the thickness of the lines, as those are barely legible in the current version;

**Reply**: Done.

**Reviewer Point P 1.7** — - ll. 226-227: I do not understand the author's choice to generate noise that has the parameters of an hypothetical multi-model mean noise. Particularly, it is not clear to me why the statistics of the noise including model would would be more representative of the metrics relation with the warming than the actual model if a model-dependent noise is applied;

**Reply**: To clarify, it's not the noise characteristics of the multi-model mean - it's the parameters for variance and lag-1 autocovariance are computed for each member of the ensemble, and our noise model uses the central estimate for those parameters. Using the actual noise from the CMIP models was considered but in practise creates more uncertainties and subjective decisions - how should internal variability be separated from forced response and long term drift? An AR1 acting as a stochastic component for weather noise in the simple model with the same order of magnitude internal variability as we see in GCMs was chosen as a simple sensitivity to a well defined source of noise. I appreciate, of course, that there is certainly an assumption of AR1 noise being representative of climate noise - which I already discuss, but I would argue that the current model is informative as a simple sensitivity test for how additive noise impacts the model fit.

**Reviewer Point P 1.8** — - ll- 281-283: the point that I made in my previous revision is not that there is an indefinite number of exponentials that can be combined to describe the response function. Rather, I wanted to stress that other methods, as those based on the Ruelle's response theory (cfr. Ragone et al. 2016, Lucarini et al. 2017, Lembo et al. 2019; I believe that the correct reference here is "Lembo V., Lucarini V., and Ragone F., 2019, Beyond Forcing Scenarios: Predicting Climate Change through Response Operators in a Coupled General Circulation Model, ArXiv, 1912.03996"), allow to extract a (linear) response function without making any assumption on the parameters describing its optimal fit. Consequent to the theory, it can be proven (see Lucarini, V., 2018, Revising and extending the linear response theory for statistical mechanical systems: Evaluating observables as predictors and predictands) that any Green function can be expressed as an infinite sum of exponentials.

**Reply**: Thanks for this clarification, I've incorporated your central point (that a general linear response function can be described with a decaying exponential basis set) in the new discussion on the motivation for the 2 timescale model in Section 1.1

**0.1 Technical comments**

**Reviewer Point P 1.9** — - l. 28: replace "Response at" with "Response at time of";

**Reply**: Done

**Reviewer Point P 1.10** — - l. 33: replace "given the" with "given that the";

**Reply**: Done

**Reviewer Point P 1.11** — - l. 73: insert blank space before "(Smith";

**Reply**: Done

**Reviewer Point P 1.12** — - l. 112: replace "inefficient requires" with "inefficient, as it requires";

**Reply**: Done

**Reviewer Point P 1.13** — - l. 250: replace "given we" with "given that we";

**Reply**: Done
* * *
**Reviewer 2**

**Reviewer Point P 2.1** — The author significantly improved the paper. A detailed and comprehensive description of the model and the optimization method has been included, which substitutes the former, in my view insufficient, appendix. Now, the setup of the model and the optimization procedure defining the parameter distribution (i.e. the model ensemble) becomes much clearer. An attempt is made to evaluate/validate the model by applying it to historical CMIP simulations and projections. The results indicate that, in general, the model performs well. In addition, it is shown that considering the drift can indeed improve (at least a little) the estimate of TCR (and other metrics). Both tests add more confidence to the main conclusions. Furthermore, the author now discusses the potential effect of the assumption of two timescale on the results, and gives some references to the linear response theory, which helps to assess the results. Overall, I believe the manuscript now provides sufficient new and valuable information to warrant publication. I only have very few minor/technical points:

**Reply**: Many thanks for the constructive review, and positive assessment of the revised study

**Reviewer Point P 2.2** — - Abstract: The acronyms ECS and TCR are used without previous definition.

**Reply**: Thanks - fixed.

**Reviewer Point P 2.3** — - L93: $F_{otherAnt} -> F_{other}$

**Reply**: Fixed

**Reviewer Point P 2.4** — - Citations: it seems that sometimes the citation styles '(authors, year)' and 'authors (year)' are mixed-up (e.g. L90, L110)

**Reply**: fixed throughout

**1 Changes in manuscript**

I have endeavored to address the minor issues brought up by reviewers in the second round of review:

- Added discussion in Section 1.1 describing the logic for the use of a 2 timescale model, and wider issues of how pulse-response models relate to generic linear response models

- corrected latex citation issues

- reformatted Eq 2

- moved figure S3 to the main text (new Figure 3), and added more extensive discussion of posterior distribution of the MCMC. Included analytical solution for TCR to provide insight into how constrained transient warming would impact parameter constraints.

- fixed minor technical issues

[revised manuscript text omitted]

**Figure S2.** Scatterplots of 21st century warming (difference between 20 year means in 2081-2100 and 1981-2000) and a range of sensitivity metrics for CMIP5. TCR, T140 and EffCS are reported values from Stocker et al. (2013). A140 is calculated as the year 131-150 average global mean temperature above the control level (taken as the last 100 years of the relevant control simulation). Columns represent different RCPs, rows represent different sensitivity metrics considered in the text. Each point represents a single model from the archive. Only results from the 1st initial condition ensemble member are considered for each model (thus the plots are subject to initial condition variability).

[Figure]

A 'corner-plot' showing the posterior parameter distribution attained by MCMC calibration of the simple climate model. Diagonal plots show posterior histograms for parameter values optimized in the calibration, while the horizontal range indicates the bounding values of the initial flat prior distribution. Off-diagonal plots show pairwise distributions of parameters in the posterior distribution.

**Figure S3.** A demonstration of the simple model fitting strategy applied to historical simulations for a range of models in the CMIP5 archive. A pulse-response model is fitted treating each model's global mean temperature output in turn as truth for the period 1870-2019 (black line). 10th-90th percentiles of fitted temperature response for historical (grey area) and future projections are shown for RCP8.5 (pink area) and RCP2.6 (blue area) concentration pathways. Dotted lines show the median temperature in the ensemble projection, while solid colored lines show the evolution of the actual GCM for the corresponding scenario.

[Figure]

**Figure S4.** Figure illustrating the 'correction' employed for TCR and ECS in Figure 6. Corrected baseline temperatures are estimated by regression of the first 20 years of the control simulation, and branch-point from the control simulation is identifying by finding the year in which a linear fit to the control model evolution intersects the corrected baseline temperature. Branching in cases where there is no intersection are illustrated by the year in which the trendline is closest to the corrected baseline (either the first or last year).

[Figure]

**Figure S5.** A figure illustrating the log of the sum squared error in best-fitting of the first 140 years of the abrupt4xCO2 time series in global mean temperature for each available model in the CMIP5 ensemble, as a function of the number of exponential modes allowed in the pulse response model. Each point shows the error in fitting one model in the CMIP5 ensemble, with colored lines tracking the error in the fitting as a function of number of allowed modes.

[Figure]

---

## Author Response (AR3)

Dear Prof. Lucarini,

Many thanks for the final comments and for the consideration of my paper.

In this version, I have addressed the issues with references, added more complete references throughout and made minor text edits for readability.

Kind regards,

Ben

[revised manuscript text omitted]

335   shows the range of fitted trend lines consistent with (a).

Scatterplots of 21st century warming (difference between 20 year means in 2081-2100 and 1981-2000) and a range of sensitivity metrics for CMIP5. TCR, T140 and EffCS are reported values from IPCC (2013), A140 is calculated as the year 131-150 average global mean temperature above the control level (taken as the last 100 years of the relevant control simulation). Columns represent different RCPs, rows represent different sensitivity metrics considered in the text. Each point represents a

340   single model from the archive. Only results from the 1st initial condition ensemble member are considered for each model (thus the plots are subject to initial condition variability).

A demonstration of the simple model fitting strategy applied to historical simulations for a range of models in the CMIP5 archive. A pulse-response model is fitted treating each model's global mean temperature output in turn as truth for the period 1870-2019 (black line). 10th-90th percentiles of fitted temperature response for historical (grey area) and future projections are

345   shown for RCP8.5 (pink area) and RCP2.6 (blue area) concentration pathways. Dotted lines show the median temperature in the ensemble projection, while solid colored lines show the evolution of the actual GCM for the corresponding scenario.

Figure illustrating the 'correction' employed for TCR and ECS in Figure 6. Corrected baseline temperatures are estimated by regression of the first 20 years of the control simulation, and branch-point from the control simulation is identifying by finding the year in which a linear fit to the control model evolution intersects the corrected baseline temperature. Branching in

350   cases where there is no intersection are illustrated by the year in which the trendline is closest to the corrected baseline (either the first or last year).

A figure illustrating the log of the sum squared error in best-fitting of the first 140 years of the abrupt4xCO2 time series in global mean temperature for each available model in the CMIP5 ensemble, as a function of the number of exponential modes

allowed in the pulse response model. Each point shows the error in fitting one model in the CMIP5 ensemble, with colored lines tracking the error in the fitting as a function of number of allowed modes.